# On Structured State-Space Duality

**Jerry Yao-Chieh Hu** [* 1 2]  **Xiwen Zhang** [* 3]  **Ali ElSheikh** [1]  **Weimin Wu** [1]  **Han Liu** [1 4]

## Abstract

Structured $\underline{S}$tate-$\underline{S}$pace $\underline{D}$uality (SSD) [Dao & Gu, ICML 2024] is an equivalence between a simple Structured $\underline{S}$tate-$\underline{S}$pace $\underline{M}$odel (SSM) and a masked attention mechanism. In particular, a state-space model with a scalar-times-identity state matrix is equivalent to a masked self-attention with a 1-semiseparable causal mask. Consequently, the same sequence transformation (model) has two algorithmic realizations: as a linear-time $O(T)$ recurrence or as a quadratic-time $O(T^2)$ attention. In this note, we formalize and generalize this duality: (i) we extend SSD from the scalar-identity case to general diagonal SSMs (diagonal state matrices); (ii) we show that these diagonal SSMs match the scalar case's training complexity lower bounds while supporting richer dynamics; (iii) we establish a necessary and sufficient condition under which an SSM is equivalent to 1-semiseparable masked attention; and (iv) we show that such duality fails to extend to standard softmax attention due to rank explosion. Together, these results tighten the bridge between recurrent SSMs and Transformers, and widen the design space for expressive yet efficient sequence models. Code is available at https://github.com/MAGICS-LAB/state_space_daulity.

[1]Center for Foundation Models and Generative AI & Department of Computer Science, Northwestern University, Evanston, IL 60208, USA [2]Ensemble AI, San Francisco, CA 94133, USA [3]University of California, San Diego, La Jolla, CA 92093, USA [4]Department of Statistics and Data Science, Northwestern University, Evanston, IL 60208, USA. Correspondence to: Jerry Yao-Chieh Hu <jhu@u.northwestern.edu>, Xiwen Zhang <xiz284@ucsd.edu>, Ali ElSheikh <alielsheikh2025@u.northwestern.edu>, Weimin Wu <wwm@u.northwestern.edu>, Han Liu <han-liu@northwestern.edu>.

*Proceedings of the 43$^{rd}$ International Conference on Machine Learning*, Seoul, South Korea. PMLR 306, 2026. Copyright 2026 by the author(s).

## 1  Introduction

Structured $\underline{S}$tate-$\underline{S}$pace $\underline{D}$uality (SSD) refers to a one-to-one equivalence between a certain linear Structured $\underline{S}$tate-$\underline{S}$pace $\underline{M}$odel (SSM) and a masked self-attention mechanism (Dao and Gu, 2024). In plain terms, it means the same sequence transformation has two algorithmic realizations: either as a recurrent state-space system or as a self-attention (matrix) operation. Dao and Gu (2024) introduce the first example of such duality: a state-space model whose state matrix is a scalar multiple of the identity is equivalent to a causal self-attention with a rank-1 mask matrix.

Concretely, consider a time-varying linear SSM driven by an input $x_t$, with hidden state $h_t$, transition matrix $A^t$, input-to-state vector $b_t$, and state-to-output vector $c_t$:

$$h_t = A^t h_{t-1} + b_t x_t, \quad y_t = c_t^\top h_t, \quad h_0 = 0.$$

Here the superscript $t$ indexes time. Unroll the recurrence

$$y_t = \sum_{s=1}^t c_t^\top A^t A^{t-1} \cdots A^{s+1} b_s x_s.$$

Define the induced sequence kernel $M \in \mathbb{R}^{T \times T}$ by

$$M_{t,s} := \begin{cases} c_t^\top A^t A^{t-1} \cdots A^{s+1} b_s, & s \le t, \\ 0, & s > t. \end{cases}$$

Then the recurrence is equivalent to the matrix map

$$Y = MX,$$

for $X = [x_1; \ldots; x_T]$ and $Y = [y_1; \ldots; y_T]$. In the scalar-identity SSD case, $A^t = a_t I_N$, so

$$M_{t,s} = \left( \prod_{r=s+1}^t a_r \right) c_t^\top b_s, \quad s \le t,$$

with the empty product equal to one so that the diagonal entries satisfy $M_{t,t} = c_t^\top b_t$. Hence the induced sequence kernel factors as

$$M = L \odot (CB^\top),$$

where the $t$-th rows of $B, C \in \mathbb{R}^{T \times N}$ are $b_t^\top$ and $c_t^\top$, and

$$L_{t,s} = \begin{cases} \prod_{r=s+1}^t a_r, & s \le t, \\ 0, & s > t. \end{cases}$$

Thus $L$ is a causal 1-semiseparable mask (see Definition 2.2

for precise definition), while $CB^\top$ is a rank-at-most-$N$ content matrix. Consequently,

$$Y = MX = (L \odot (CB^\top))X.$$

Namely, the SSM and the masked attention realize the same sequence-to-sequence map $X \mapsto Y$: one via a linear-time $O(T)$ (SSM-)recurrence and the other via a quadratic-time $O(T^2)$ (attention-)matrix multiplication.

Remarkably, this duality bridges two disparate paradigms for sequence modeling — recurrent state-space models (Dao and Gu, 2024; Gu and Dao, 2023) and Transformer attention (Vaswani et al., 2017). State-space models update a latent state recurrently, and hence yields linear complexity in sequence length. Attention mechanisms compute pairwise token interactions, and hence yields quadratic complexity. The structured state-space duality unifies these two paradigms by revealing that they implement identical functions.

Nevertheless, while Dao and Gu (2024) suggest that analogues duality should hold for diagonal state-space models, a formal treatment in the SSM-attention setting appears to be missing. We provide a self-contained proof in the SSM-attention language considered here.[1] Building on it, we formalize diagonal structured state-space duality and give an explicit computation algorithm.

**Contributions.** In this work, we build on SSD and extend its scope in four key directions:

- **General Diagonal SSMs (Sections 3.1 and 3.2).** We extend SSD beyond the simple (scalar) $\times I_N$ state matrix to general diagonal state matrices. This enlarges the class of SSMs under the duality (from a single exponential decay to $N$ separate diagonal dynamics), thereby supporting richer sequence dynamics.[2]

- **Efficiency at Scale (Section 3.3).** We prove that these more general diagonal SSMs match the training complexity lower bound of the scalar case while offering greater expressiveness. In other words, it is possible to train and execute the richer diagonal SSMs with the same optimal $O(TNd)$ time complexity as the scalar SSM, so we get additional modeling power at no extra asymptotic cost.

- **Higher-Rank Equivalence (Section 3.4).** We complement (Dao and Gu, 2024, Proposition 3.4) with a

self-contained, constructive, strictly causal proof. It establishes the equivalence between (i) lower-triangular matrices of semiseparable rank at most $N$ and (ii) lower-triangular matrices admitting an $N$-dimensional sequential state-space representation, in the SSM-attention setting considered here. While the underlying equivalence is known in the structured-matrix (quasiseparable) literature (Eidelman et al., 2014, Chapter 5), our formulation is tailored to causal SSM kernels and is reusable in subsequent duality results.

- **Masked Attention Duality of General SSM (Section 3.4).** While each 1-semiseparable masked attention has an SSM dual, we provide a necessary and sufficient condition for an $N$-semiseparable (Definition 2.1) matrix (corresponding to an SSM of state dimension $N$) to have a 1-semiseparable masked attention dual.

Together, these results strengthen the bridge between recurrent state-space models and Transformer-style attention mechanisms. They also widen the design space for expressive yet efficient sequence modeling. Through this generalized duality framework, we enable principled exploration of new architectures that enjoy the best of both worlds (recurrent and attention).

**Organization.** We provide related work in Appendix A, and background in Section 2. We show our main theory in Section 3, and the limitations of structured state-space duality in Section 4. Appendix C validates our theory with numerical studies.

**Notations.** We denote the index set $\{1, \ldots, I\}$ by $[I]$. We denote vectors with lower case and matrices with upper case. We write the sequence (time) index of a matrix as a superscript ($A^t$). We write the sequence (time) index of a vector or scalar as a subscript ($a_t$). For a matrix $A$, $A_{i,j}$ denotes its entry on the $i$-th row and the $j$-th column. $A_{i,:}$ denotes its $i$-th row, $A_{:,j}$ denotes its $j$-th column. $A_{i_1:i_2,j_1:j_2} \in \mathbb{R}^{(i_2-i_1)\times(j_2-j_1)}$ denotes its submatrix containing all its entries $A_{i,j}$ where $i_1 \le i \le i_2 - 1$ and $j_1 \le j \le j_2 - 1$. We use $\odot$ for Hadamard multiplication. We write the input sequence as $X = [x_1; \ldots; x_T] \in \mathbb{R}^{T\times d}$, where $x_t \in \mathbb{R}^{1\times d}$, and similarly $Y = [y_1; \ldots; y_T] \in \mathbb{R}^{T\times d}$.

## 2 Background

In this section, we present the ideas we build on. Then we review the prior results of state-space duality (SSD) from (Dao and Gu, 2024) for the structured state-space duality in Section 2.2.

### 2.1 Semiseparable Matrices

To prepare our theory, we begin by defining the class of semiseparable (SS) matrices.

$N$-**Semiseparable ($N$-SS) Matrix.** We give a definition of

---

[1] Similar equivalences are known in the structured-matrix (quasiseparable) literature (Eidelman et al., 2014, Chapter 5), but they are not formulated in the SSM-attention language considered here.

[2] For example, a diagonal SSM with two distinct decay rates can simultaneously capture both fast and slow time-scale patterns in a sequence, something a single-decay (scalar) SSM cannot do. In a toy regression task with a mixture of two exponential signals, a 2-dimensional SSM can fit both modes, whereas any 1-dimensional SSM fails (Appendix C.7).

this matrix whose lower-triangular submatrices have union-bounded rank as follows.

**Definition 2.1** ($N$-Semiseparable ($N$-SS) Matrix). A lower triangular matrix $M$ is $N$-*semiseparable* (N-SS) if every submatrix of $M$ consisting of entries on or below the main diagonal has rank at most $N$. The smallest such $N$ is called the semiseparable *rank* (or order) of $M$.

In essence, an $N$-semiseparable matrix allows up to $N$ independent directions in each lower-triangular block.

1**-Semiseparable (1-SS Matrix).** Now we define a special case of $N$-SS matrix with very low rank: 1-semiseparable matrix.

**Definition 2.2** (1-Semiseparable (1-SS) Matrix). Suppose $M$ is a lower triangular matrix. $M$ is *1-semiseparable* (1-SS) if and only if every submatrix of $M$ consisting of entries on or below the main diagonal has rank at most 1.

Intuitively, a 1-SS matrix has extremely low complexity: each new row introduces at most one new independent direction in the space of lower-triangular entries.

**Remark 2.1** (Prior Work). We remark that, 1-SS is the common setup when considering SS matrix analysis (Fasino and Gemignani, 2002; Vandebril et al., 2005; 2008). In contrast, we consider SS with higher rank.

We next define a 1-SS masked attention operator. In essence, it is an attention layer with 1-SS matrix as its causal mask.

**Definition 2.3** (1-SS Masked Attention). Let $Q, K \in \mathbb{R}^{T \times N}$ be query and key matrices. A *1-SS masked attention* is a self-attention operation whose attention weight matrix is masked by a 1-SS matrix $L \in \mathbb{R}^{T \times T}$. In particular, the attention scores take the form $L \odot (QK^\top)$, i.e. the element-wise product of $QK^\top$ with the mask $L$ (with $L$ enforcing a causal lower-triangular structure).

Furthermore, we introduce the notion of a *Sequentially SemiSeparable* (SSS) representation. Importantly, SSS connects these structured matrices to state-space models:

**Definition 2.4** ($N$-Sequentially Semiseparable ($N$-SSS) Representation). A lower triangular matrix $M \in \mathbb{R}^{T \times T}$ has an $N$-*sequentially semiseparable (N-SSS) representation* if there exist vectors $b_1, \ldots, b_T \in \mathbb{R}^N$, $c_1, \ldots, c_T \in \mathbb{R}^N$, and matrices $A^1, \ldots, A^T \in \mathbb{R}^{N \times N}$ such that

$$M_{j,i} = c_j^\top A^j \cdots A^{i+1} b_i, \qquad (2.1)$$

for all $1 \leq i \leq j \leq T$.

**Definition 2.5** ($N$-SSS Representable Matrix). A matrix $M$ is $N$-*SSS representable* if it admits an $N$-SSS representation (i.e., (2.1)). Equivalently, $M$ can be written in the form of (2.1).

The above definitions formalize how a structured state-space model of dimension $N$ gives rise to a matrix $M$ with semiseparable rank $N$. In particular, any $M$ that has an $N$-SSS representation is necessarily $N$-semiseparable (since each new state dimension contributes at most one new rank to the growing matrix). We next review the known correspondence between such structured matrices and attention mechanisms in the simplest (rank-1) case.

## 2.2 Structured State-Space Duality (SSD)

We now describe the structured state-space duality as introduced by Dao and Gu (2024) for the scalar-identity state-space case. We begin by formulating the state-space model and its induced sequence kernel, then show how it corresponds to a masked attention operator.

**Time-Varying SSM and Induced Kernel.** Consider a time-varying linear state-space model (SSM) with state dimension $N$, defined by the recurrence

$$\underbrace{h_t}_{N \times d} := \underbrace{A^t}_{N \times N} \underbrace{h_{t-1}}_{N \times d} + \underbrace{b_t}_{N \times 1} \underbrace{x_t}_{1 \times d}, \ \underbrace{y_t}_{1 \times d} = \underbrace{c_t^\top}_{1 \times N} \underbrace{h_t}_{N \times d}, \ t \in [T],$$

(2.2)

where we set $h_0 = 0$ for consistency. Here, $h_t \in \mathbb{R}^{N \times d}$ is the hidden state, $A^t \in \mathbb{R}^{N \times N}$ is the state transition matrix, $b_t \in \mathbb{R}^{N \times 1}$ and $c_t \in \mathbb{R}^{N \times 1}$ are input and output weight matrices. Importantly, this recurrence defines a causal linear operator on the input sequence. Unrolling the recurrence yields an explicit input-output relation

$$y_t = \sum_{s=1}^t M_{t,s} x_s, \ M_{t,s} := \begin{cases} c_t^\top A^t \cdots A^{s+1} b_s, & t \geq s; \\ 0, & t < s. \end{cases}$$

(2.3)

for $1 \leq s \leq t \leq T$. We refer to $M_{t,s}$ as the *SSM kernel* at $(t, s)$. Let $M \in \mathbb{R}^{T \times T}$ denote the lower-triangular matrix of kernel coefficients, i.e. $M_{t,s}$ for $t \geq s$ (and $M_{t,s} = 0$ for $t < s$). By construction, $M$ encodes the entire transformation from inputs $x_1, \ldots, x_T$ to outputs $y_1, \ldots, y_T$. Moreover, $M$ is structured: since the latent state is $N$-dimensional, $M$ has semiseparable rank at most $N$ (each row of $M$ lies in an $N$-dimensional subspace). In particular, $M$ is an $N$-SS matrix in the sense of Definition 2.1, and for this special case $N = 1$, $M$ is 1-SS.

**Scalar-Times-Identity State Matrix** ($A^t = a_t I_N$). A simple case of the above is when each state matrix is a scalar multiple of the identity. We call such an SSM a *scalar-identity SSM*, meaning $A_t = a_t I_N$ for some scalar $a_t \in \mathbb{R}$. In this case, the recurrence (2.2) simplifies to

$$y_t = \sum_{s=1}^t a_t \cdots a_{s+1} c_t^\top b_s x_s, \qquad (2.4)$$

which is a convolution-style sum over past inputs. For example, if $a_t = a$ is constant, then (2.4) reduces to the standard discrete-time convolution $y_t = \sum_{s=1}^{t} a^{t-s} c_t^\top b_s x_s$.

**Scalar-Identity SSM.** We call an SSM layer a scalar-identity SSM if each of the state matrices $A^t$ is a scale multiple of the identity matrix.

**Rank-1 Special Case ($N = 1$).** In the extreme case of state dimension $N = 1$, the state $h_t$ is one-dimensional. Then $b_t$ and $c_t$ are scalars for all $t$. We collect the input sequence into a matrix $X = [x_1; x_2; \ldots; x_T] \in \mathbb{R}^{T \times d}$ (with $x_t^\top$ as the $t$-th row) and similarly $Y = [y_1; \ldots; y_T] \in \mathbb{R}^{T \times d}$ for the outputs. Let $p = (b_1, \ldots, b_T)^\top \in \mathbb{R}^T$ and $q = (c_1, \ldots, c_T)^\top \in \mathbb{R}^T$ denote the vectors of input and output weights over time. The scalar-identity formula (2.4) then reduces to

$$\underbrace{Y}_{T \times d} = \underbrace{\mathrm{diag}(q)}_{T \times T} \underbrace{M}_{T \times T} \underbrace{\mathrm{diag}(p)}_{T \times T} \underbrace{X}_{T \times d},$$

where

$$M_{t,s} := \begin{cases} a_t \cdots a_{s+1}, & \text{for} \quad t \ge s; \\ 0, & \text{for} \quad t < s. \end{cases}$$

Here $M$ is a 1-semiseparable mask matrix (Definition 2.2). Hence $\mathrm{diag}(q) M \mathrm{diag}(p) = M \odot (qp^\top) = M \odot (CB^\top)$, i.e., $M$ serves as a causal mask on the outer-product matrix $CB^\top = qp^\top$. In other words, the sequence mapping implemented by this $N = 1$ SSM can be viewed as a 1-SS masked attention operation with $Q = C$ and $K = B$.

Now we are ready to present the structured state-space duality by Dao and Gu (2024):

---

**Proposition 2.1** (Dao and Gu (2024) Scalar-Identity State-Space Duality). Consider the SSM defined by (2.2) where each $A^t = a_t I_N$ (i.e. a scalar-identity SSM). Let $B = [b_1; b_2; \ldots; b_T]^\top \in \mathbb{R}^{T \times N}$ and $C = [c_1; c_2; \ldots; c_T]^\top \in \mathbb{R}^{T \times N}$ be the matrices whose $t$-th rows are $b_t^\top$ and $c_t^\top$, respectively. Define $M \in \mathbb{R}^{T \times T}$ by $M_{t,s} = a_t a_{t-1} \cdots a_{s+1}$ for $t \ge s$ and $M_{t,s} = 0$ for $t < s$. Then for any input sequence $X = [x_1; \ldots; x_T] \in \mathbb{R}^{T \times d}$ with output $Y = [y_1; \ldots; y_T] \in \mathbb{R}^{T \times d}$, the recurrence (2.4) is equivalent to a 1-SS masked attention representation:

$$Y = (M \odot (CB^\top))X,$$

where

$$M_{t,s} := \begin{cases} a_t \cdots a_{s+1}, & \text{for} \quad t \ge s; \\ 0, & \text{for} \quad t < s. \end{cases}$$

Here $\odot$ denotes elementwise (Hadamard) product. In particular, the same sequence transformation is realizable either by the linear-time recurrence (2.2) or by the quadratic-time matrix operation on the right-hand side.

---

Proposition 2.1 (Dao and Gu, 2024, Prop. 3.4) establishes a one-to-one correspondence between a simple structured SSM and a masked self-attention operator with a 1-SS (rank-1) mask.

## 3 Main Theory

This section presents our main results: (i) we extend the state-space duality to general diagonal SSMs in Section 3.1; (ii) we specify the case of state-space duality for diagonal SSMs with full-rank state matrices in Section 3.2; (iii) we give a concrete algorithm for computing diagonal SSD layer and show that the computational complexity of diagonal SSD is the same as scalar-identity SSD in Section 3.3; and (iv) we find the exact class of general SSMs that have 1-SS masked attention dual in Section 3.4.

### 3.1 Structured State-Space Duality for General Diagonal SSMs

Dao and Gu (2024) study state-space duality of SSM with scalar-identity state matrices. Here we complement (Dao and Gu, 2024) by extending state-space duality to SSM with general diagonal state matrices. In the case of general diagonal SSM, each state matrix $A^t$ is a diagonal matrix. The diagonal state-space model has an attention-like dual as the sum of $N$ attention matrices.

**Attention-Like Dual of Diagonal SSMs.** Suppose $M \in \mathbb{R}^{T \times T}$ is a lower triangular matrix as in (2.1) regarding the state-space model. We show that $M$ has an attention-like dual as the sum of $N$ attention-like matrices

$$M^n = L^n \odot (Q^n \cdot K^{n\top}),$$

where for all $n \in [N]$ we have $Q^n, K^n \in \mathbb{R}^{T \times 1}$.

Specifically, suppose

$$M_{j,i} = c_j^\top A^j \cdots A^{i+1} b_i, \quad \text{for all} \quad 1 \le i \le j \le T, \tag{3.1}$$

where $b_1, \cdots, b_T, c_1, \cdots, c_T \in \mathbb{R}^N$ and each $A^t \in \mathbb{R}^{N \times N}$ is a diagonal matrix. Please see Figure 1 for a visualization.

Then we have

$$M_{j,i} = \sum_{n=1}^{N} (c_j)_n (A^j \cdots A^{i+1})_{n,n} (b_i)_n,$$

for all $1 \le i \le j \le T$.

Note that we separate those terms for different $n_1, n_2 \in [N]$. For all $n \in [N]$, let $b^n, c^n \in \mathbb{R}^T$ be such that for all $t \in [T]$, $b_t^n = (b_t)_n$ and $c_t^n = (c_t)_n$. Please see Figure 2 for a visualization.

Define $1\mathsf{SS}(\cdot) : \mathbb{R}^T \to \mathbb{R}^{T \times T}$ by

$$1\mathsf{SS}(a_1, a_2, \cdots, a_T) = \underbrace{M}_{T \times T},$$

$$\mathbf{M_{j,i}} = \mathbf{c_j^\top A^j \cdots A^{i+1} b_i}$$

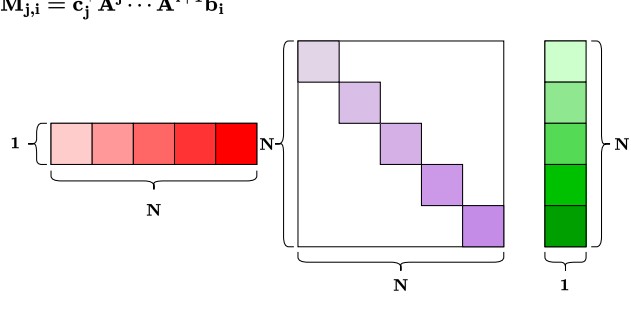

$$\mathbf{c_j^\top \in \mathbb{R}^{1 \times N}} \qquad \mathbf{A^j \cdots A^{i+1} \in \mathbb{R}^{N \times N}} \qquad \mathbf{b_i \in \mathbb{R}^N}$$

*Figure 1.* $M_{j,i} = c_j^\top A^j \cdots A^{i+1} b_i$

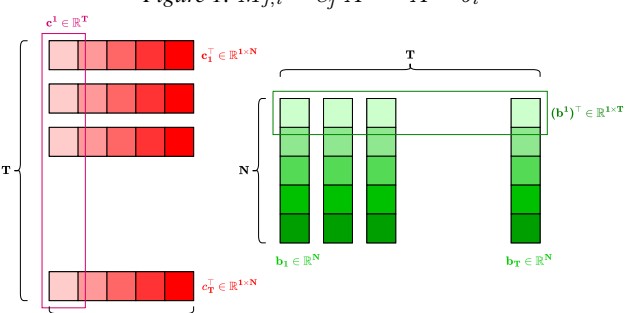

*Figure 2.* Construction of $b^n$ and $c^n$.

where for $1 \le t, s \le T$ it holds that

$$M_{t,s} := \begin{cases} a_t \cdots a_{s+1}, & \text{for} \quad t \ge s; \\ 0, & \text{for} \quad t < s. \end{cases}$$

Then for all $n \in [N]$, we consider the 1-SS masked attention matrix $M^n$ as

$$M^n = \mathsf{1SS}(A_{n,n}^1, \cdots, A_{n,n}^T) \odot (c^n \cdot b^{n\top}).$$

We have $M = \sum_{n=1}^N M^n$ by simple algebra.

So far we construct the attention-like dual for general SSM. It enables us to further construct the 1-SS masked attention dual for a specific class of diagonal SSM.

### 3.2 Structured State-Space Duality for Diagonal SSMs with Full-Rank State Matrices

Next we show that, when all the state matrices $A^t$ of the state-space model have full rank, the attention-like dual of diagonal SSM turns into 1-SS masked attention dual. Specifically, we use the attention-like representation of $M$ in Section 3.1 to construct the 1-SS masked attention dual of $M$.

**1-SS Attention Dual of SSM with Full-Rank Diagonal State Matrices.** Suppose $M \in \mathbb{R}^{T \times T}$ has $N$-SSS representation as in (2.1), where each $A^t$ is a diagonal matrix with

none-zero determinant. In this case we show that $M$ has a 1-SS masked attention dual.

Specifically, when $\det(A^t) \ne 0$ for all $t \in [T]$, $M^n$ has the representation of (for $n \in [N]$)

$$\mathsf{1SS}(1, 1, \cdots, 1) \odot (c'^n \cdot b'^{n\top}),$$

where

$$c_t'^n := c_t^n \cdot (A_{n,n}^1 \cdots A_{n,n}^t), \quad b_t'^n := \frac{b_t^n}{A_{n,n}^1 \cdots A_{n,n}^t},$$

for all $t \in [T]$. Then, we let $B', C' \in \mathbb{R}^{T \times N}$ be such that $B'_{:,n} = b'^n$ and $C'_{:,n} = c'^n$ for all $n \in [N]$, we have

$$M = \mathsf{1SS}(1, 1, \cdots, 1) \odot (C' \cdot B'^\top).$$

In this way we construct the 1-SS masked attention dual of diagonal SSM with full-rank state matrices.

**Remark 3.1** (An Example of "Richer Dynamics" of Diagonal SSM). The analysis above shows substantial overlap between the kernels (equivalently, masked-attention matrices) representable by scalar-identity SSMs and by diagonal SSMs. However, diagonal SSMs are strictly more expressive: there exist kernels realizable by a diagonal SSM of state dimension $N$ that cannot be realized by any scalar-identity SSM with the same $N$. For example, let $N = 2$, $T = 4$, and consider

$$M = \begin{bmatrix} 2 & 0 & 0 & 0 \\ 1 & 2 & 0 & 0 \\ 0 & 1 & 2 & 0 \\ 0 & 0 & 1 & 2 \end{bmatrix} = \begin{bmatrix} 1 & 0 & 0 & 0 \\ 1 & 1 & 0 & 0 \\ 0 & 0 & 1 & 0 \\ 0 & 0 & 1 & 1 \end{bmatrix} + \begin{bmatrix} 1 & 0 & 0 & 0 \\ 0 & 1 & 0 & 0 \\ 0 & 1 & 1 & 0 \\ 0 & 0 & 0 & 1 \end{bmatrix}.$$

This decomposition exhibits $M$ as a sum of two 1-SS components, hence $M$ is realizable by a diagonal SSM with state dimension $N = 2$ (equivalently, as a sum of two 1-SS heads). In contrast, $M$ has $3 > 2 = N$ new columns, so it cannot be realized by any scalar-identity SSM with state dimension $N = 2$. In Appendix C.7, we further validate this "richer dynamics" phenomenon via multivariate time-series regression experiments.

### 3.3 Computational Complexity of Diagonal SSD

We give the concrete computation algorithm of diagonal state-space duality according to the attention-like dual in Section 3.1 and evaluate its efficiency in three aspects: (i) computation cost; (ii) total memory; (iii) parallelization.

**Computation Algorithm of Diagonal SSD.** We define the functions $f : \mathbb{R}^T \times \mathbb{R}^{T \times d} \to \mathbb{R}^{T \times d}$ by

$$f(x, Y)_{:,s} = x \odot (Y_{:,s}), \quad \text{for all} \quad s \in [d],$$

and $g : \mathbb{R}^T \times \mathbb{R}^{T \times d} \to \mathbb{R}^{T \times d}$ by

$$g(x, Y)_{1,:} = Y_{1,:},$$

and

$$g(x, Y)_{t+1,:} = x_{t+1} \cdot g(x, Y)_{t,:} + Y_{t+1,:},$$

for $t \in [T-1]$. Consider the SSM layer with state dimension $N$ defined by (2.2), where each $A^t$ is a diagonal matrix. The recurrence relation (2.2) also has representation

$$\underbrace{Y}_{T \times d} = \underbrace{M}_{T \times T} \cdot \underbrace{X}_{T \times d},$$

where

$$M_{j,i} = c_j^\top A^j \cdots A^{i+1} b_i.$$

Express $M$ as $M = \sum_{n=1}^N M^n$, where

$$M^n = \mathsf{1SS}(A_{n,n}^1, \cdots, A_{n,n}^T) \odot (c^n \cdot b^{n\top}).$$

Let $a^n \in \mathbb{R}^T$ denote $(A_{n,n}^1, \cdots, A_{n,n}^T)$ for $n \in [N]$. Denote $Y = M \cdot X$ as $Y = \mathrm{SSM}(X)$. Then $\mathrm{SSM}(X)$ is computed as the following algorithm Algorithm 1.

---

**Algorithm 1** Diagonal State-Space Dual (SSD)

---
**procedure** SSM($X$)
$\quad Z^n \leftarrow f(b^n, X) \quad$ for all $n \in [N] \quad$ // Time $\ O(NTd)$
$\quad H^n \leftarrow g(a^n, Z^n) \quad$ for all $n \in [N]$ // Time $\ O(NTd)$
$\quad Y^n \leftarrow f(c^n, H^n) \quad$ for all $n \in [N]$ // Time $\ O(NTd)$
$\quad Y \leftarrow \sum_{n=1}^N Y^n \quad\quad\quad\quad\quad\quad$ // Time $\ O(NTd)$
$\quad$ **return** $Y$
**end procedure**

---

**Computation Cost Same as Scalar-Identity SSD.** Since each step of Algorithm 1 takes computation cost of $O(NTd)$, this algorithm takes total computation cost of $O(NTd)$ FLOPs. This implies diagonal SSD has the same computation cost as scalar-identity SSD (Dao and Gu, 2024, Theorem 3.8), and matches the maximum parameter complexity for SSD (Dao and Gu, 2024, Appendix A.1).

**Remark 3.2.** Specifically, we compute the $O(NTd)$ term as $NTd$ flops in each step and $4NTd$ flops in total. Therefore our algorithm not only has the same FLOP complexity order as SSD, but also exhibits a similar constant factor.

**Total Memory Cost.** The memory cost of the state data $A^1, \cdots, A^T, b_1, \cdots, b_T, c_1, \cdots, c_T$ is $TN + TN + TN = O(NT)$. In the first three steps of Algorithm 1, each step generates $N$ matrices of size $T \times d$, and in the last step of Algorithm 1, only one matrix of size $T \times d$ is generated. Therefore the memory cost of the intermediate step is $NTd + NTd + NTd + Td = O(NTd)$. Considering all the memory costs above, we deduce that diagonal state-space duality has total memory cost $O(NT) + O(NTd) = O(NTd)$.

**Parallelization.** Note that the first three steps of Algorithm 1 are operated in parallel for each $n \in [N]$, the diagonal state-space dual has separation into $N$ parallel computation processes, each of them costing time $O(Td)$. Furthermore,

from the definition of $f$ and $g$, note that each column of $X$ operates respectively during the whole processing of Algorithm 1. Therefore the diagonal state-space dual has further separation into $Nd$ parallel computation processes, each process costing time $O(T)$.

**Remark 3.3.** At present, there is no specialized kernel that directly accelerates diagonal-SSD computations. This is not a fundamental limitation: from a hardware perspective, diagonal SSDs admit the same asymptotic time complexity and parallelization structure as scalar-identity SSMs, and can in principle be implemented with similar accelerator support. Realizing such performance requires dedicated diagonal-SSD kernel design, which we leave to future work. The focus of this paper is to characterize the duality itself, rather than to develop optimized implementations.

### 3.4 General SSMs with 1-SS Masked Attention Duals

We further study the duality between 1-SS masked attention and general SSM, and provide the exact class of general SSM that has 1-SS masked attention dual. Firstly, we state the equivalence between the class of $N$-SS matrices and the class of $N$-SSS representable matrices. In particular, we identify a direct 1-1 correspondence between SSMs and $N$-SSS representations. Thus, each SSM has a corresponding $N$-SS matrix. Then we study the duality between 1-SS masked attention and general SSM regarding the SSM's corresponding $N$-SS matrix.

**Equivalence Between $N$-SS Matrices and $N$-SSS Representable Matrices.** We start with a constructive proof for an equivalence introduced in (Dao and Gu, 2024).

**Proposition 3.1** (Proposition 3.3 in (Dao and Gu, 2024))**.** A lower triangular matrix is $N$-semiseparable iff it is $N$-SSS representable.

*Proof.* Please see Appendix B.1 for a detailed proof. $\qquad\square$

**Remark 3.4.** We remark that Proposition 3.1 complements the proof of (Dao and Gu, 2024, Proposition 3.3). This constructive proof forms a rigorous basis for general SSD and yields an explicit procedure to obtain the $N$-SSS representation from an $N$-SS matrix. However, its computational complexity exceeds SSD limit (i.e., Section 3.3), and fast algorithms for more general SSD are still lacking. We leave the design of such fast algorithms to future work.

**Remark 3.5.** The equivalence between 1-SS and 1-SSS-representable matrices appears in structured-matrix literature such as semiseparable, quasiseparable, and SSS forms (Vandebril et al., 2005). We extend it to $N$-SS matrices. The equivalence between semiseparable matrices and structured state-space representations has been studied in the structured-matrix literature (e.g., under the notion of quasiseparable matrices). Our contribution here is not to claim

a new mathematical equivalence, but to provide a self-contained, constructive, and strictly causal proof formulated entirely in the SSM/SSD notation. This formulation aligns directly with state-space models and masked attention, and is what enables us to use the equivalence as a building block in later results (e.g., Theorem 3.1 and the diagonal SSD).

So far we obtain the equivalence between the class of $N$-SS matrices and the class of $N$-SSS-representable matrices. Moreover, we have a direct one-to-one correspondence between $N$-SSS representation and general SSM. Next, we use $N$-SS matrices to study the duality between 1-SS masked attention and general SSM.

**SSMs with 1-SS Masked Attention Dual.** Here, we provide a necessary and sufficient condition for an SSM to have 1-SS masked attention dual. In particular, we state this condition in terms of the SSM's corresponding $N$-SS matrix. Then, we characterize the possible attention dual for this $N$-SS matrix. We start with a few concepts below for convenience of discussion:

**Definition 3.1** (Fine 1-SS Matrix). We say a 1-SS matrix $L = 1\mathsf{SS}(a_1, a_2, \cdots, a_t)$ is a *fine* 1-SS matrix iff $a_1 a_2 \cdots a_t \neq 0$.

**Definition 3.2** (New Column of Lower Triangular Matrix). Suppose $M \in \mathbb{R}^{T \times T}$ is a lower triangular matrix. We call $M_{t:,t}$ a *new column* of $M$ iff $M_{t:T+1,t}$ is not in $M_{t:,:t}$'s column space.

Intuitively, a "new column" of $M$ is a column that is not in the span of the preceding columns. Equivalently, it introduces a new direction in the column space that cannot be synthesized from earlier columns. In the state-space view, each new column corresponds to an additional independent "memory direction" that must be carried by the latent state, i.e., information not representable by the previously accumulated state.

We firstly specify the class of $N$-SS matrices that has a representation of fine 1-SS masked attention. This prepares us for the later general State-Space Duality in Theorem 3.1.

**Proposition 3.2.** Suppose $M \in \mathbb{R}^{T \times T}$ is an $N$-SS lower triangular matrix. Then $M$ has representation of 1-SS masked attention $L \odot (QK^\top)$ for some $Q, K \in \mathbb{R}^{T \times N}$ and fine 1-SS matrix $L$ iff it has at most $N$ new columns.

*Proof.* Please see Appendix B.2 for detailed proof. □

Then we deduce the following results from Proposition 3.2. These results state the necessary and sufficient condition for an SSM to have a general 1-SS masked attention dual.

**Lemma 3.1.** Suppose $M \in \mathbb{R}^{T \times T}$ is an $N$-SS lower-triangular matrix. Then $M$ admits a 1-SS masked-attention

representation

$$M = L \odot (QK^\top), \quad Q, K \in \mathbb{R}^{T \times N},$$

for some 1-SS mask $L$ if and only if there exist indices

$$1 = t_0 < t_1 < \cdots < t_R = T + 1,$$

such that, writing

$$M^{(r)} := M_{t_{r-1}:t_r, t_{r-1}:t_r}, \quad r \in [R],$$

the following two conditions hold:

- all nonzero entries of $M$ lie in the contiguous principal blocks $M^{(1)}, \ldots, M^{(R)}$,

- each block $M^{(r)}$ has at most $N$ new columns.

Here new columns are counted within each block, after viewing $M^{(r)}$ as a lower-triangular matrix.

*Proof.* The proof consists of two parts.

**Part I: Sufficient Condition.** Here, we show that if $M$ has contiguous principal blocks containing all nonzero entries of $M$, and each block has at most $N$ new columns, then $M$ has a 1-SS masked-attention representation.

Suppose $M \in \mathbb{R}^{T \times T}$ has $R$ contiguous principal blocks

$$M_{t_0:t_1, t_0:t_1}, M_{t_1:t_2, t_1:t_2}, \ldots, M_{t_{R-1}:t_R, t_{R-1}:t_R},$$

where

$$1 = t_0 < t_1 < \cdots < t_R = T + 1.$$

Denote the $r$-th block by

$$M^{(r)} := M_{t_{r-1}:t_r, t_{r-1}:t_r}, \quad r \in [R].$$

Then $M^{(r)}$ has size $(t_r - t_{r-1}) \times (t_r - t_{r-1})$.

By Proposition 3.2, each block has a representation

$$M^{(r)} = 1\mathsf{SS}(\underbrace{1, \ldots, 1}_{t_r - t_{r-1}}) \odot (Q^{(r)} K^{(r)\top}),$$

for some $Q^{(r)}, K^{(r)} \in \mathbb{R}^{(t_r - t_{r-1}) \times N}$.

Then, let

$$p^{(r)} := (0, 1, \ldots, 1) \in \mathbb{R}^{t_r - t_{r-1}}, \quad r \in [R],$$

and concatenate

$$p^\top := [p^{(1)\top}, p^{(2)\top}, \ldots, p^{(R)\top}] \in \mathbb{R}^{1 \times T},$$
$$Q^\top := [Q^{(1)\top}, Q^{(2)\top}, \ldots, Q^{(R)\top}] \in \mathbb{R}^{N \times T},$$
$$K^\top := [K^{(1)\top}, K^{(2)\top}, \ldots, K^{(R)\top}] \in \mathbb{R}^{N \times T}.$$

The zeros in $p$ make all cross-block entries vanish, while the restriction of $1\mathsf{SS}(p)$ to each block is the all-ones causal mask. Since all nonzero entries of $M$ lie inside these blocks,

we have

$$M = 1\mathsf{SS}(p) \odot (QK^\top).$$

**Part 2: Necessary Condition.** Here, we show that if $M$ has a 1-SS masked-attention representation, then $M$ has contiguous principal blocks containing all its nonzero entries, and each block has at most $N$ new columns.

Suppose

$$M = 1\mathsf{SS}(p) \odot (QK^\top),$$

for some $p \in \mathbb{R}^T$ and $Q, K \in \mathbb{R}^{T \times N}$. Let $t_1, \ldots, t_{R-1}$ be the zero positions among $p_2, \ldots, p_T$, and set

$$t_0 = 1, \quad t_R = T + 1.$$

Then all nonzero entries of $M$ lie in the contiguous principal blocks

$$M_{t_0:t_1,t_0:t_1}, \quad M_{t_1:t_2,t_1:t_2}, \quad \ldots, \quad M_{t_{R-1}:t_R,t_{R-1}:t_R}.$$

Indeed, if an entry crosses a boundary $t_r$, then the corresponding $1\mathsf{SS}(p)$ product contains the zero factor $p_{t_r}$.

For each block, we have

$$M_{t_{r-1}:t_r,t_{r-1}:t_r} = 1\mathsf{SS}(\widetilde{p}^{(r)}) \odot (Q_{t_{r-1}:t_r,:} K_{t_{r-1}:t_r,:}^\top),$$

for all $r \in [R]$, where

$$\widetilde{p}^{(r)} := (1, p_{t_{r-1}+1}, \ldots, p_{t_r-1}) \in \mathbb{R}^{t_r - t_{r-1}}.$$

The first parameter of $1\mathsf{SS}(\cdot)$ does not affect the mask, and all other entries of $\widetilde{p}^{(r)}$ are nonzero by construction. Hence each block has a fine 1-SS masked-attention representation. By Proposition 3.2, each block has at most $N$ new columns.

This completes the proof. $\qquad\square$

So far we characterize the exact class of $N$-SS matrices corresponding to a 1-SS masked attention representation. Given the equivalence between $N$-SS matrices and $N$-SSS representable matrices, along with the direct one-to-one correspondence between $N$-SSS-representable matrices and SSMs, we obtain the following theorem:

> **Theorem 3.1** (General State-Space Duality). Suppose $M \in \mathbb{R}^{T \times T}$ is the $N$-semiseparable kernel corresponding to an SSM. Then this SSM has a 1-SS masked-attention dual if and only if there exist indices $1 = t_0 < t_1 < \cdots < t_k = T + 1$ such that all nonzero entries of $M$ lie in the contiguous principal blocks $M_{t_{i-1}:t_i,;t_{i-1}:t_i}$, and each block has at most $N$ new columns.

Each new column represents one additional independent memory direction. Thus, Theorem 3.1 says that an exact 1-SS dual exists precisely when no active block requires more than the $N$ memory directions available to the state. See Remark 3.8 for the structural interpretation. Figure 3 gives a visual summary of this condition.

**Remark 3.6.** In Proposition 3.2, we focus on fine 1-SS masked attention, whereas Lemma 3.1 handles general 1-SS masked attention. In the case of general 1-SS masked attention, $a_t$ is possibly 0 for some $t \in [T]$ in the causal mask $L = 1\mathsf{SS}(a_1, a_2, \cdots, a_T)$.

**Remark 3.7.** Mathematically, Lemma 3.1 follows from Proposition 3.2. Proposition 3.2 establishes the structural form for the fine 1-SS case (Definition 3.1). The general 1-SS case can be obtained by decomposing the matrix into several diagonal blocks, each of which is a fine 1-SS matrix. Applying Proposition 3.2 to each block immediately yields Lemma 3.1.

**Remark 3.8** (Structural Nature of 1-SS Duality). Theorem 3.1 is a structural theorem: it gives a maximally general necessary-and-sufficient condition for when an SSM admits a 1-SS masked-attention dual. Since it applies to arbitrary SSMs (independent of parameterization, training objective, or optimization method) the statement is necessarily abstract and should not be read as a training-time recipe. Our proof is constructive in the mathematical sense: it identifies which subspaces (directions) must be preserved or can be merged to obtain a 1-SS representation. The goal is to characterize the solution set, not to prescribe an implementation. Turning these structural insights into practical, hardware-efficient algorithms is an important direction for future work.

# 4 Limitations of Structured State-Space Duality

In this section, we study the limitations of state-space duality from its two ends: (i) on the attention side, we show the impossibility of extending SSD to practical softmax attention (Vaswani et al., 2017); (ii) on the SSM side, we show that even SSMs with low state dimension or low-rank state matrices may not always admit an SSD extension.

**Impossibility of Extending SSD to Softmax Attention.** We present a simple example showing that softmax attention (Vaswani et al., 2017) does not admit state-space duality. Consider a matrix $V \in \mathbb{R}^{T \times T}$ with entries $V_{i,j} = i \times j$ for all $i, j \in [T]$. The matrix $V$ has rank 1 since every column is a scalar multiple of $(1, 2, \ldots, T)^\top \in \mathbb{R}^T$. However, $\mathrm{Softmax}(V)$ has rank $T$ by the Vandermonde determinant, and in fact, every submatrix of $\mathrm{Softmax}(V)$ is full rank.

This shows that even when the attention matrix $QK^\top$ has very low rank, $\mathrm{Softmax}(QK^\top)$ typically expands to full rank $T$. Moreover, any attention matrix with a state-space dual must be $N$-semiseparable, where $N$ is the state dimension of the corresponding SSM. Hence, softmax attention does not have a state-space model dual.

**Impossibility of Extending SSD to General SSM with Low State Dimension.** We provide an example for this impossibility in the following proposition. It shows that a

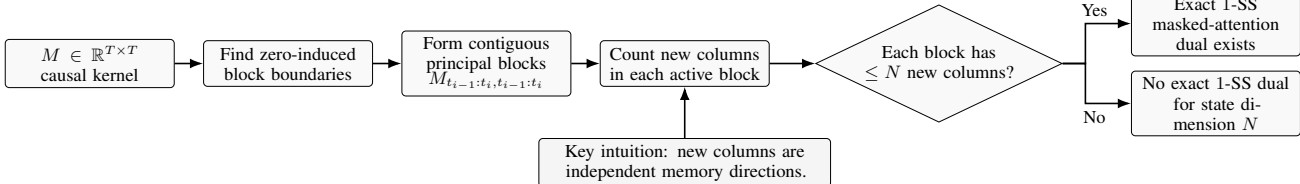

*Figure 3.* **Theorem 3.1 Intuition.** Zeros in the 1-SS mask partition the kernel into contiguous principal blocks. Within each active block, the number of new columns equals the number of independent memory directions required by the state. Hence, an exact 1-SS dual exists if and only if every active block requires at most $N$ such directions.

general SSM, *even* with low state dimension or low-rank state matrices, *may not* admit a 1-SS masked-attention dual.

**Proposition 4.1.** Consider the SSM layer with state dimension $N \geq 2$ defined by (2.2), there exist $A^{1:T+1}, b_{1:T+1}, c_{1:T+1}$ such that the recurrence relation does not have an attention dual.

*Proof.* According to Proposition 3.1, there exist $A^{1:T+1}, b_{1:T+1}, c_{1:T+1}$ such that the recurrence relation (2.2) has representation $Y = M \cdot X$, where $M := I_{T \times T} + E^{T,1}$ is a 2-SS matrix. Here

$$E_{j,i}^{T,1} := \begin{cases} 1, & j = T \text{ and } i = 1; \\ 0, & \text{otherwise.} \end{cases}$$

We claim that $M$ does not have the representation of $L \odot (QK^\top)$ where $Q, K \in \mathbb{R}^{T \times N}$ and $L$ is a 1-SS matrix.

Otherwise if $M = L \odot (QK^\top)$, suppose

$$L_{j,i} = \begin{cases} a_j \cdots a_{i+1}, & \text{for} \quad j \geq i; \\ 0, & \text{for} \quad j < i. \end{cases}$$

for $i, j \in [T]$.

Since $M_{T,1} = 1$, $L_{T,1} = a_2 \cdots a_T$ is none-zero, i.e. each of $a_2, a_3, \cdots, a_T$ is none-zero.

Thus, we have

$$(QK^\top)_{i,j} = 0, \quad \text{for all} \quad 1 \leq j < i \leq T - 1.$$

Since each diagonal element of $M$ is none-zero, each diagonal element of $QK^\top$ is also none-zero. Given that $(QK^\top)_{i,j} = 0$ for all $1 \leq j < i \leq T - 1$, we deduce that $QK^\top$ has rank at least $T - 1$. This is a contradiction to $\text{rank}(QK^\top) \leq \text{rank}(Q) \leq N$.

This completes the proof. $\qquad\square$

Therefore, general SSM does not always have 1-SS masked attention dual, even when the SSM has very low state dimension.

**Remark 4.1.** Our rank-explosion obstruction is consistent with (Sieber et al., 2024). In their dynamical-systems formulation, an exact recurrent realization of softmax attention requires unbounded state dimension. In our terms,

for generic $Q, K$, the matrix $\text{Softmax}(QK^\top)$ becomes (numerically) full rank for large $T$ (not $N$-SS for any fixed $N$), so no finite-state SSM dual exists. Therefore, no finite-state SSM (equivalently, no fixed-size RNN) can represent softmax self-attention exactly.

# 5 Conclusion

Structured state-space models (SSMs) (Dao and Gu, 2024; Gu and Dao, 2023) and attention mechanisms (Vaswani et al., 2017) represent two contrasting paradigms for sequence modeling. SSMs evolve a latent state step by step and run in linear time $O(T)$. Attention mixes all pairs and costs $O(T^2)$. Structured State-Space Duality (SSD) (Dao and Gu, 2024) provides a theoretical bridge between these two views. We formalize and generalize the structured state-space duality between simple recurrent state-space models (SSMs) and masked attention mechanisms.

Specifically, we extend the duality introduced by (Dao and Gu, 2024) from the scalar-identity case to general diagonal SSMs (Section 3.1) and show that these models retain the same computational complexity lower bounds while supporting richer dynamics (Section 3.3). We further characterize the necessary and sufficient condition under which an SSM corresponds to a 1-semiseparable masked attention mechanism (Section 3.4). We also present two negative results (Section 4): (i) this duality does not extend to standard softmax attention due to a rank explosion in the induced kernel; (ii) even for low-rank SSMs, such duality may still fail at representational level. Lastly, we validate our results with numerical experiments (Appendix C). Together, these results strengthen the theoretical bridge between recurrent SSMs and Transformer-style attention, and broaden the design space for expressive yet efficient sequence modeling.

## Impact Statement

By the theoretical nature of this work, we do not anticipate any negative social impact.

## Acknowledgments

JH thank Alex Reneau, Mimi Gallagher, Sara Sanchez, T.Y. Ball, Dino Feng and Andrew Chen for enlightening discussions on related topics, and the Red Maple Family for

support. The authors would like to thank the anonymous reviewers and program chairs for constructive comments.

JH is partially supported by Ensemble AI, and the Walter P. Murphy Fellowship and the Terminal Year Fellowship (Paul K. Richter Memorial Award) of Northwestern University. Han Liu is partially supported by NIH R01LM1372201, NSF AST-2421845, Simons Foundation MPS-AI-00010513, AbbVie , Dolby and Chan Zuckerberg Biohub Chicago Spoke Award. This research was supported in part through the computational resources and staff contributions provided for the Quest high performance computing facility at Northwestern University which is jointly supported by the Office of the Provost, the Office for Research, and Northwestern University Information Technology. The content is solely the responsibility of the authors and does not necessarily represent the official views of the funding agencies.

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

# Appendix

## LLM Usage Disclosure

We used large language models (LLMs) to aid and polish writing, such as improving clarity, grammar, and conciseness. We also used LLMs for retrieval and discovery, for example exhausting literature to identify potential missing related work. All technical content, proofs, experiments, and results are original contributions by the authors.

## A  Related Work

In this section, we review related work on structured state-space models, efficient Transformers and linear attention mechanisms, and the connections between state-space models and attention.

**Structured State-Space Models (SSMs).**    SSMs aim to model long-range dependencies with linear-time computation. S3 first introduced the idea of parameterizing state-space models with structured matrices to enable efficient sequence modeling (Gu et al., 2021). S4 introduces a state-space layer with stable diagonalization and fast convolution, which enabled long-context training and inference (Gu et al., 2021). S5 simplifies the design with a single multi-input multi-output SSM and preserved $O(T)$ scaling (Smith et al., 2023). Mamba adds input-dependent gating on the SSM projections and achieved strong accuracy with linear-time sequence modeling (Gu and Dao, 2023). These works establish SSMs as competitive sequence models and motivate analyses that compare their expressive power to attention.

**Efficient Transformers and Linear Attention.**    Many works reduce the quadratic cost of self-attention by imposing structure or approximation (Tay et al., 2022). Linformer projects keys and values to low rank and reduced compute and memory while preserving accuracy (Wang et al., 2020). Linear Transformers replace softmax by kernel feature maps and execute attention as a recurrence, which yielded $O(T)$ autoregressive inference (Katharopoulos et al., 2020). Performer uses random features to approximate softmax attention with variance control and linear complexity (Choromanski et al., 2021). Nyströmformer applies Nyström approximation to attention and obtains sub-quadratic cost (Xiong et al., 2021). Sparse patterns such as Longformer, BigBird, and Reformer improve scaling by local windows, global tokens, or LSH-based routing (Beltagy et al., 2020; Zaheer et al., 2020; Kitaev et al., 2020). Retentive networks propose a recurrent retention operator that matches Transformer quality with linear-time execution (Sun et al., 2023). These methods show that attention admits efficient surrogates when the attention matrix has a low-rank, kernel, or sparse structure.

**Connections between SSMs and Attention.**    Linear attention admits a recurrent implementation and thus links attention with RNN-style computation (Katharopoulos et al., 2020). Dao and Gu (2024) introduce *Structured State Duality* (SSD), prove the duality between scalar-identity SSMs and masked attention with 1-semiseparable kernels, and conjectrue such duality hold in diagonal SSM. Hydra generalizes matrix mixers beyond causal SSMs with quasi-separable structure and bidirectional information flow (Hwang et al., 2024).

Our work differs in scope and goal: we give an algebraic duality between $N$-dimensional diagonal SSM and 1-semiseparable masked attention, and we prove that diagonal SSMs match the scalar case in training FLOPs and memory while enabling $N$

independent state modes. These results place diagonal SSD on a firm foundation and clarifies how SSM capacity aligns with semiseparable rank.

# B Proof of Main Text

## B.1 Equivalence between N-SS Matrices and N-SSS-Representable Matrices

**Proposition B.1** (Proposition 3.3 in (Dao and Gu, 2024); Proposition 3.1 Restate). A lower triangular matrix $M$ is $N$-semiseparable iff it is $N$-SSS-representable, i.e. $M$ being $N$-SSS representable is the necessary and sufficient condition for $M$ to be $N$-semiseparable.

*Proof.* Our proof consists of two parts.

**Part 1: Sufficient Conditon.** In this part, we take three steps to show that any lower triangular matrix with an $N$-SSS representation is $N$-semiseparable.

- **Step 1: Express $M$ with its $N$-SSS representation.** Suppose $M \in \mathbb{R}^{T \times T}$ is a lower triangular matrix with an $N$-SSS representation

$$M_{j,i} = c_j^\top A_j \cdots A_{i+1} b_i, \tag{B.1}$$

  for all $i, j \in [T]$, where $b_1, \cdots, b_T, c_1, \cdots, c_T \in \mathbb{R}^N$ and $A^1, \cdots, A^T \in \mathbb{R}^{N \times N}$.

- **Step 2: Express each submatrix $S$ on or below the diagonal of $M$ through the $N$-SSS representation of $M$.** Suppose $S$ is a submatrix of $M$ such that each entry of $S$ is on or below the diagonal line of $M$, we have

$$S = M_{j_1:j_2, i_1:i_2},$$

  for some $1 \le j_1 < j_2 \le T + 1$, $1 \le i_1 < i_2 \le T + 1$ and $i_2 \le j_1$.

  Denote $S^1 \in \mathbb{R}^{(j_2 - j_1) \times N}$ as

$$S^1[:, j] = c_{j+j_1-1}^\top A^{(j+j_1-1)} \cdots A^{j_1+1},$$

  for $j \in [j_2 - j_1]$.

  Denote $S^2 \in \mathbb{R}^{N \times (i_2 - i_1)}$ as

$$S^2[i, :] = A^{j_1} \cdots A^{(i+i_1)} b_{i+i_1-1}$$

  for $i \in [i_2 - i_1]$.

  Then according to (B.1), we have

$$S = S_1 \cdot S_2.$$

- **Step 3: Upperbound the rank of $S$ with $S^1$ and $S^2$.** Since $S^1$ and $S^2$ both have rank at most $N$, we deduce that $S$ has rank at most $N$.

**Part 2: Neccessary Condition.** In this part we take four steps to show that any $N$-semiseparable lower triangular matrix has an $N$-SSS representation. Suppose $M \in \mathbb{R}^{T \times T}$ is an $N$-semiseparable lower triangular matrix.

- **Step 1: Divide $M_{t:,:t+1}$ into the product of two matrices of rank at most $N$.** In this step, we represent $M_{t:,:t+1}$ by the product of two low-rank matrices for all $t \in [T]$.

  Since $M$ is $N$-semiseparable, $M_{t:,:t+1} \in \mathbb{R}^{(T-t+1) \times t}$ has rank at most $N$ for each $t \in [T]$. Therefore there exist low-rank matrices $W_t \in R^{(T-t+1) \times N}$ and $U_t \in \mathbb{R}^{N \times t}$ such that

$$M_{t:,:t+1} = W_t \cdot U_t. \tag{B.2}$$

  Please see Figure 4 for a visualization.

  Let $r_t \le N$ denote the rank of $M_{t:,:t+1}$ for all $t \in [T]$. Without loss of generality, we construct $W_t$ and $U_t$ to be such that only the first $r_t$ columns of $W_t$ are none-zero, and similarly, only the first $r_t$ rows of $U_t$ are none-zero. Please see Figure 5 for a visualization.

  This means that the span of $W_t$'s column vectors equals the span of $M_{t:T+1,1:t+1}$'s column vectors, and the span of $U^t$'s row vectors equals the span of $M_{t:T+1,1:t+1}$'s row vectors.

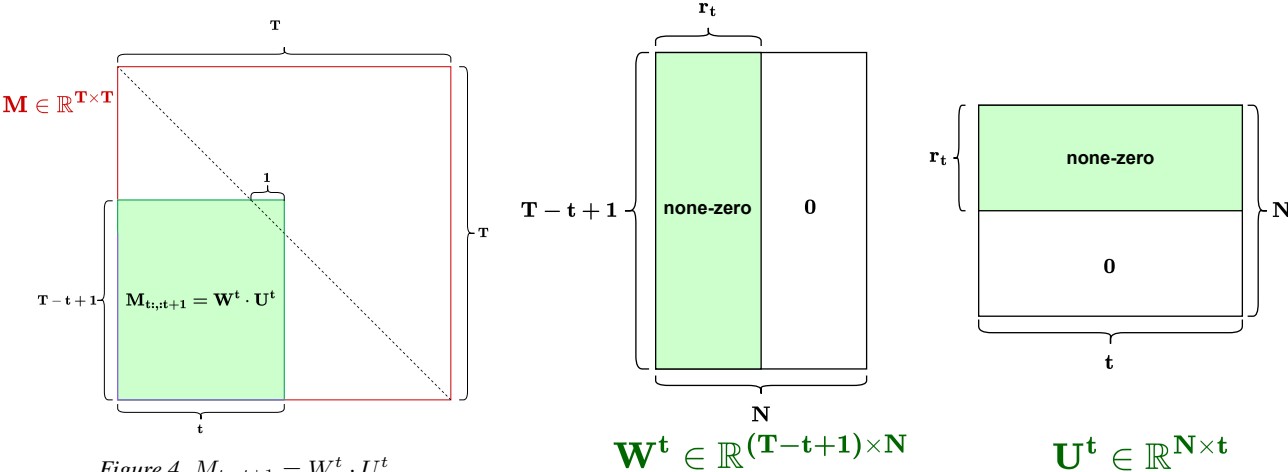

Figure 4. $M_{t:,:t+1} = W^t \cdot U^t$

Figure 5. Only the first $r_t$ columns (rows) of $W^t$ ($U^t$) are non-zero.

- **Step 2: Conditions on $A^t, b_t, c_t$ when $M$ has an $N$-SSS representation.** In this step, we characterize the conditions that the state parameters $A^{1:T+1}, b_{1:T+1}, c_{1:T+1}$ should satisfy when $M$ has an $N$-SSS representation.

  We start by defining $W^{t'}$ and $U^{t'}$ for convenience of discussion.

  Set $W^{t'} = W^t_{2:,:}$ and $U^{t'} = U^t_{:,:t}$ for all $t \in [T]$, i.e. $W^{t'}$ is $W^t$ without the first row, and $U^{t'}$ is $U^t$ without the last column. We provide a visualization in Figure 6.

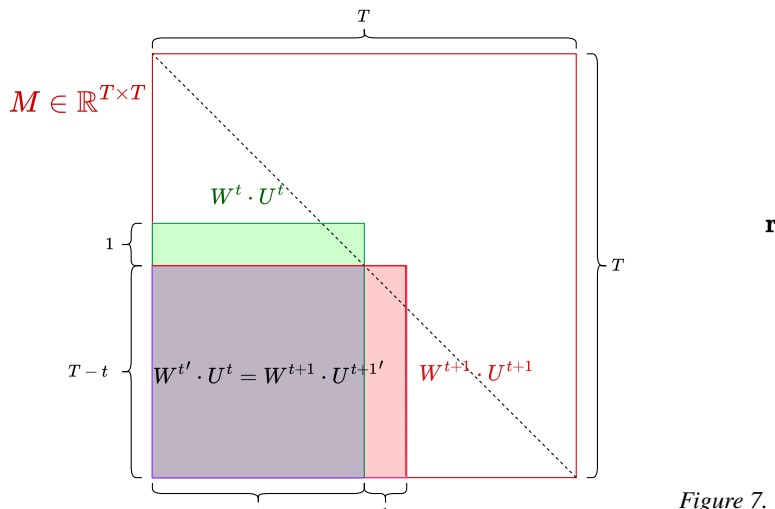

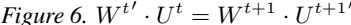

Figure 6. $W^{t'} \cdot U^t = W^{t+1} \cdot U^{t+1'}$

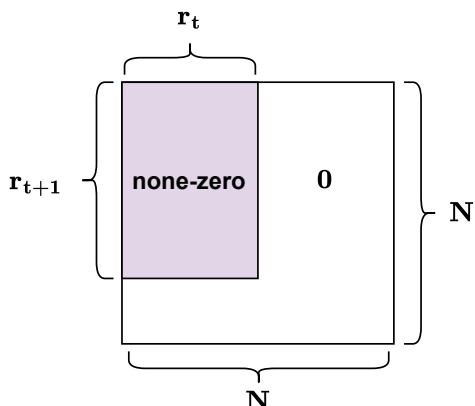

Figure 7. Set only the first $r_t$ columns and the first $r_{t+1}$ rows of $A^{t+1}$ and $A^{t+1'}$ to be non-zero.

Then we have

$$W^{t+1} \cdot U^{t+1'} = M_{t+1:,:t+1} = W^{t'} \cdot U^t. \tag{B.3}$$

Note that if $M$ has an $N$-SSS representation, then for all $t \in [T]$, we have

$$M_{t:,:t+1} = \begin{pmatrix} c_t^\top \\ c_{t+1}^\top A^{t+1} \\ c_{t+2}^\top A^{t+2} A^{t+1} \\ \vdots \\ c_T^\top A^T \cdots A^{t+1} \end{pmatrix} \cdot \begin{pmatrix} A^t \cdots A^2 b_1 & A^t \cdots A^3 b_2 & \cdots & A^t b_{t-1} & b_t \end{pmatrix}. \tag{B.4}$$

We expect there exist $A^{1:T+1}$, $b_{1:T+1}$, $c_{1:T+1}$ satisfying

$$\begin{pmatrix} c_t^\top \\ c_{t+1}^\top A^{t+1} \\ c_{t+2}^\top A^{t+2} A^{t+1} \\ \vdots \\ c_T^\top A^T \cdots A^{t+1} \end{pmatrix} = W^t, \tag{B.5}$$

and

$$\begin{pmatrix} A^t \cdots A^2 b_1 & A^t \cdots A^3 b_2 & \cdots & A^t b_{t-1} & b_t \end{pmatrix} = U^t, \tag{B.6}$$

for all $t \in [T]$.

We observe that (B.5) and (B.6) require $W^{t'} = W^{t+1} \cdot A^{t+1}$ and $U^{t+1'} = A^{t+1} \cdot U^t$ for all $t \in [T-1]$.

- **Step 3. Verify the existence of $A^t$ satisfying the conditions in Step 2.** In this step, we show that there exists $A^{t+1} \in \mathbb{R}^{N \times N}$ for all $t \in [T-1]$ satisfying $W^{t'} = W^{t+1} \cdot A^{t+1}$ and $U^{t+1'} = A^{t+1} \cdot U^t$.

  Since the column space of $W^{t+1}$ equals to the column space of $M_{t+1:,:t+2}$, and the column space of $M_{t+1:,:t+2}$ contains the column space of $W^{t'}$, we deduce that the column space of $W^{t+1}$ contains the column space of $W^{t'}$. Thus, there must exist $A^{t+1} \in \mathbb{R}^{N \times N}$ satisfying $W^{t'} = W^{t+1} \cdot A^{t+1}$.

  For the same reason there exists $A^{t+1'} \in \mathbb{R}^{N \times N}$ satisfying $U^{t+1'} = A^{t+1'} \cdot U^t$.

  Without loss of generality, we set only the first $r_t$ columns and the first $r_{t+1}$ rows of $A^{t+1}$ and $A^{t+1'}$ to be none-zero. Please see Figure 7 a visualization.

  Now we have

  $$\begin{aligned} W^{t+1} \cdot A^{t+1'} \cdot U^t &= W^{t+1} \cdot U^{t+1'} \\ &= M_{t+1:,:t+1} \\ &= W^{t'} \cdot U^t \\ &= W^{t+1} \cdot A^{t+1} \cdot U^t, \end{aligned}$$

  where the 1st step is by $U^{t+1'} = A^{t+1'} \cdot U^t$, the 2nd and 3rd step is by (B.3) (see Figure 6), and the last step is by $W^{t'} = W^{t+1} \cdot A^{t+1}$. We provide a visualization in Figure 8.

  Since $W^{t+1}$ and $U^t$ have rank $r_{t+1}$ and $r_t$ perspectively, we deduce that $A^{t+1'} = A^{t+1}$. So far for any $t \in [T-1]$, we construct $A^{t+1}$ satisfying both $W^{t'} = W^{t+1} \cdot A^{t+1}$ and $U^{t'} = A^{t+1'} \cdot U^{t+1}$.

- **Step 4. Construct the $N$-SSS representation of $M$.** So far we construct $A^t$, $W^t$ and $U^t$ satisfying (B.2), (B.5), and (B.6). In this step we construct the $N$-SSS representation of $M$ using the $A^t$, $W^t$ and $U^t$.

  For all $t \in [T]$, let $c_t$ be the first column of $(W^t)^\top$ and $b_t$ be the last column of $U^t$. Let $A^{t+1}$ be as constructed above for $t \in [T-1]$ and $A^1 = I_N$. Then

  $$M_{j,i} = c_j^\top A^j \cdots A^{i+1} b_i.$$

  Namely, $M$ has an $N$-SSS representation.

This completes the proof. □

## B.2  Condition for N-SS Matrix to Have Fine 1-SS Masked Attention Dual

**Proposition B.2** (Proposition 3.2 Restate).  Suppose $M \in \mathbb{R}^{T \times T}$ is an $N$-SS lower triangular matrix. Then $M$ has representation of 1-SS masked attention $L \odot (QK^\top)$ for some $Q, K \in \mathbb{R}^{T \times N}$ and fine 1-SS matrix $L$ iff it has at most $N$ new columns, i.e. $M$ having at most $N$ new columns is the necessary and sufficient condition for $M = L \odot (QK^\top)$.

*Proof.* The proof consists of two parts.

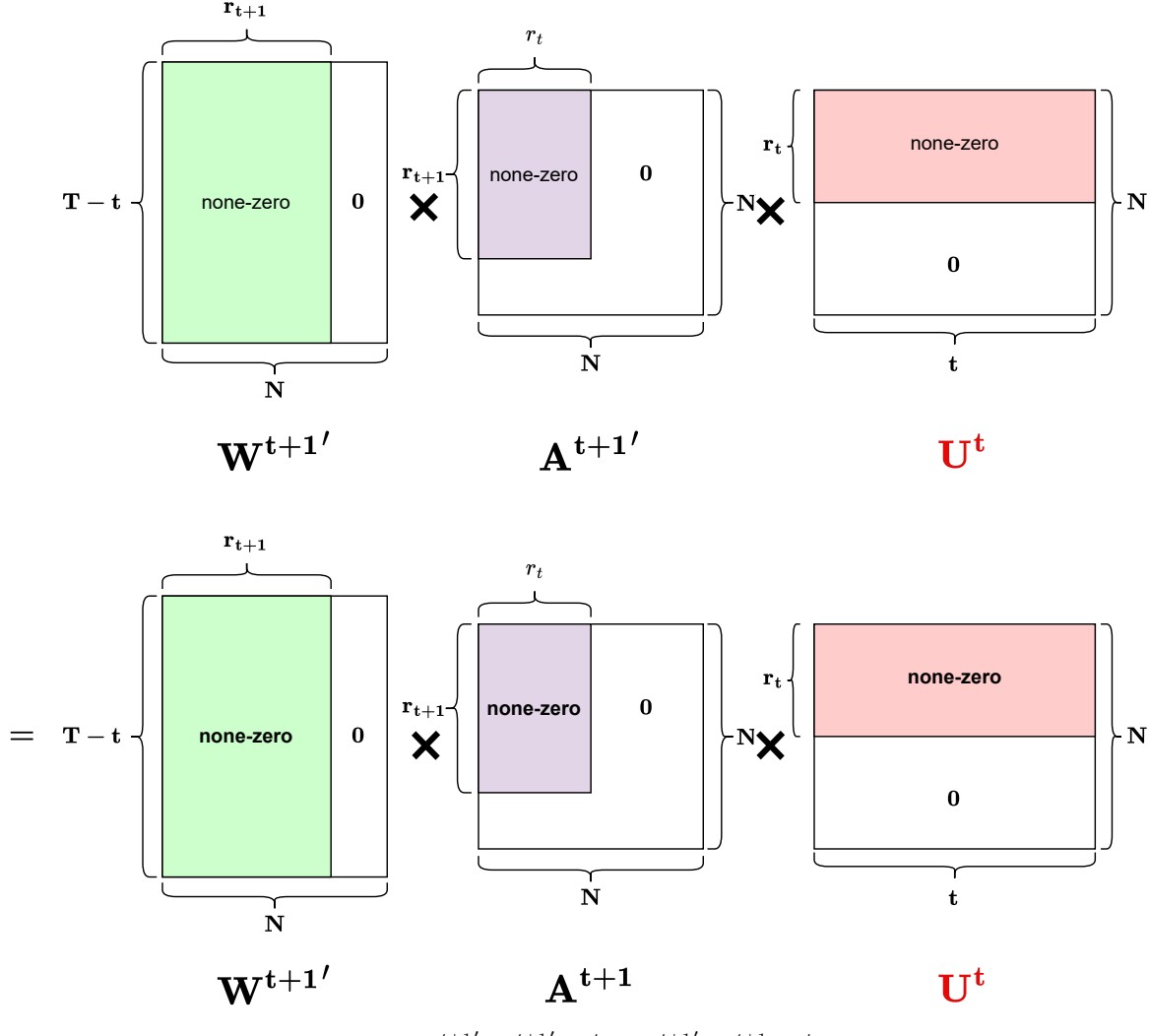

Figure 8. $W^{t+1'} \cdot A^{t+1'} \cdot U^t = W^{t+1'} \cdot A^{t+1} \cdot U^t$

**Part 1: Necessary Condition.** In this part we show that $M$ does not have representation of fine 1-SS masked attention if it has more than $N$ new columns.

Suppose

$$M = L \odot QK^\top \tag{B.7}$$

for some $Q, K \in \mathbb{R}^{T \times N}$ and $L = \mathsf{1SS}(a_1, a_2, \cdots, a_T)$ where $a_1 a_2 \cdots a_T \neq 0$, and suppose that $M$ has at least $N + 1$ new columns.

We then multiply the $t$-th row by $\frac{1}{a_1 a_2 \cdots a_t}$ and multiply the $t$-th column by $a_1 a_2 \cdots a_t$ for all $t \in [T]$. Note that these operations do not change the number of new columns of $M$.

After these operations, we get a new lower triangular matrix $M'$ having at least $N + 1$ new columns. We deduce by (B.7) that the lower triangular part of $M'$ is exactly the same as the lower triangular part of $QK^\top$. Denote $W := QK^\top$, then $W$ has rank at most $N$.

Suppose $M'$ has new columns $M'_{t_1:,t_1}, M'_{t_2:,t_2}, \cdots M'_{t_{N+1}:,t_{N+1}}$ for $1 \le t_1 < t_2 < \cdots < t_{N+1} \le T$.

We claim that $W_{:,t_1}, W_{:,t_2}, \cdots, W_{:,t_{N+1}}$ are linearly independent. If not so, there exist $c_1, c_2, \cdots, c_{N+1} \in \mathbb{R}$ such that at

least one of them is none-zero and

$$c_1 W_{:,t_1} + c_2 W_{:,t_2} + \cdots + c_{N+1} W_{:,t_{N+1}} = \mathbf{0}_T.$$

Suppose $n$ is the largest index in $[N+1]$ such that $c_{t_n} \neq 0$. Then we have

$$c_1 W_{:,t_1} + c_2 W_{:,t_2} + \cdots + c_n W_{:,t_n} = \mathbf{0}_T.$$

This implies that

$$c_1 M'_{t_n:,t_1} + c_2 M'_{t_n:,t_2} + \cdots + c_n M'_{t_n:,t_n}$$
$$= c_1 W_{t_n:,t_1} + c_2 W_{t_n:,t_2} + \cdots + c_n W_{t_n:,t_n}$$
$$= \mathbf{0}_{T-t_n+1},$$

which contradicts to the fact that $M'_{t_n:,t_n}$ is a new column of $M'$.

Therefore we deduce that $W_{:,t_1}, W_{:,t_2}, \cdots, W_{:,t_{N+1}}$ are linearly independent. This implies that $W$ has rank at least $N+1$, which contradicts to $W = QK^\top$. Thus, $M$ does not have the representation of $L \odot (QK^\top)$, where $Q, K \in \mathbb{R}^{T \times N}$ and $L$ is a fine 1-SS matrix.

**Part 2: Sufficient Condition.**    In this part we show that any lower triangular matrix with at most $N$ new columns has representation of $L \odot (QK^\top)$ for some $Q, K \in \mathbb{R}^{T \times N}$ and fine 1-SS matrix $L \in \mathbb{R}^{T \times T}$.

Suppose $M \in \mathbb{R}^{T \times T}$ is a lower triangular matrix having at most $N$ new columns. We now change $M$'s entries above the diagonal to create a matrix with rank at most $N$.

We change the entries column by column from the left to the right.

For $t \in [T]$, if $M_{t:,t}$ is a new column of $M$, remain $M'_{:t,t}$ to be $\mathbf{0}_{t-1}$; if $M_{t:,t}$ is not a new column of $M$, there exist $c_1, c_2, \cdots, c_{t-1} \in \mathbb{R}$ satisfying

$$M_{t:,t} = \sum_{s=1}^{t-1} c_s M t:,s.$$

Set $M'_{:t,t}$ to be

$$\sum_{s=1}^{t-1} c_s M'_{:t,s},$$

then $M'_{:,:t+1}$ and $M'_{:,:t}$ have the same column rank.

Given that $M$ has no more than $N$ new columns, we deduce by mathematical induction that $M'$ has rank at most $N$. Therefore there exist $Q, K \in \mathbb{R}^{T \times N}$ such that $M' = QK^\top$.

Then we have

$$M = 1\mathsf{SS}(1, 1, \cdots, 1) \odot (QK^\top).$$

This implies that $M$ has the representation of fine 1-SS masked attention matrix.

This completes the proof. □

# C  Numerical Studies

In this section, we provide empirical validation of our theoretical results across three dimensions: numerical equivalence, structural properties, and practical modeling.

- In Appendices C.1 and C.2, we verify the exact equivalence between State-Space Models (SSMs) and semiseparable attention mechanisms.

- In Appendices C.3 to C.5, we confirm the theoretical predictions regarding the rank structure and computational complexity of these kernels compared to standard softmax attention.

- In Appendix C.6, we evaluate the learning capabilities of the proposed diagonal parameterization by benchmarking an "SSD-Mamba" variant against a standard Mamba baseline on a language modeling task.

- In Appendix C.7, we use synthetic time series regression experiments to validate the efficacy of the additional modes (i.e., the richer dynamics) in the proposed diagonal SSM benchmarked against vanilla scaler SSM.

## C.1  Scalar SSM vs. 1-SS Attention Equivalence (Proposition 2.1)

To verify the fundamental connection between simple recurrences and attention mechanisms, we validate the exact equivalence between a scalar linear state-space model (SSM) and a single-head semiseparable (1-SS) masked attention layer, as stated in Proposition 2.1.

**Setup.** For a scalar input sequence $(u_t)_{t=0}^{T-1}$, we compare the SSM outputs:

$$x_t = ax_{t-1} + bu_t, \quad y_t = cx_t,$$

with the outputs of a causal attention operator with a geometric decay mask:

$$M_{t,s} = a^{t-s}\mathbb{1}\{t \geq s\}, \quad y_t^{\text{att}} = cb \sum_{s \leq t} M_{t,s} u_s.$$

The experiment outputs the maximum absolute discrepancy $(\max_t |y_t - y_t^{\text{att}}|)$ for a fixed $(a, b, c)$ and horizon $T$.

**Data.** We work with scalar sequences of length $T$. For each run, we draw an input vector $u \in \mathbb{R}^T$ with i.i.d. standard normal entries $u_t \sim N(0, 1)$ using a fixed random seed. All computations use double-precision floating point (float64).

**Model.** For the SSM, we implement the recurrence above, with $x_0 = 0$ and iterate it forward to obtain $y \in \mathbb{R}^T$. On the attention side, we construct a causal mask $\text{mask}_{t,s} = \mathbb{1}\{t \geq s\}$ together with the time/lag indices $t, s \in \{0, \ldots, T-1\}$. These form the semiseparable kernel $M_{t,s} = a^{t-s}\text{mask}_{t,s}$, which when applying the resulting linear operator to $u$, yielding $y^{\text{att}} = cbMu$.

This matches the closed-form solution of the SSM unrolled over time.

**Baseline / Ground Truth.** We treat the direct forward recurrence as the ground truth, since it defines the core dynamics of the scalar SSM. The attention implementation is an algebraic rewrite of the same linear operator.

**Results.** Across a sweep of sequence lengths $T$, decay parameters $a$, and 1000 different random seeds, the maximum absolute error $\max_t |y_t - y_t^{\text{att}}|$ remains below $10^{-14}$ in all runs. This confirms that the 1-SS masked attention operator reproduces the scalar SSM trajectory up to numerical precision. This experiment provides a concrete numerical check of the scalar SSM $\equiv$ 1-SS attention equivalence claimed in Proposition 2.1 and illustrates how a simple recurrence can be viewed as a degenerate semiseparable attention head.

## C.2  Diagonal SSM vs. Sum of Attention Heads (Section 3.1)

Moving to higher-dimensional latent spaces, we validate the structural equivalence between a diagonal SSM with state dimension $N$ and a sum of $N$ one-semiseparable (1-SS) masked attention heads, as stated in Section 3.1.

**Setup.** For a scalar input sequence, $(u_t)_{t=0}^{T-1}$, the diagonal SSM maintains an internal state $x_t \in \mathbb{R}^N$ and updates:

$$x_t = Ax_{t-1} + Bu_t, \quad y_t = C^\top x_t,$$

where $A = \mathrm{diag}(\lambda_1, \ldots, \lambda_N)$ contains the decay factors and $B, C \in \mathbb{R}^N$ are input and output weight vectors.

Unrolling this recurrence yields:

$$y_t = \sum_{n=1}^N C_n B_n \sum_{s \leq t} \lambda_n^{t-s} u_s.$$

Equivalently, we can write the same operator as the sum of $N$ 1-SS attention heads with kernels:

$$M_{t,s}^{(n)} = \lambda_n^{t-s} \mathbb{1}\{t \geq s\}, \quad M_{t,s} = \sum_{n=1}^N C_n B_n M_{t,s}^{(n)}, \quad y^{\mathrm{att}} = Mu.$$

The experiment computes the maximum absolute discrepancy $\max_t |y_t - y_t^{\mathrm{att}}|$ and compares rank of the attention kernel to the generator rank.

**Data.** We use scalar input sequences $u \in \mathbb{R}^T$ with i.i.d. entries $u_t \sim N(0,1)$ sampled from a NumPy random number generator with a fixed seed. Unless otherwise stated, we use a state dimension $N = 2$ with distinct decay rates $\lambda = (0.5, 0.8)$. We set the input/output couplings to $B = C = \mathbb{1} \in \mathbb{R}^2$. All computations are done using double precision (float64).

**Model.** For the SSM component, we implement a diagonal SSM with $N$ scalar-mode recurrence:

$$x_t^{(n)} = \lambda_n x_{t-1}^{(n)} + B_n u_t, \quad y_t = \sum_{n=1}^N C_n x_t^{(n)},$$

with $x_0 = 0$. In our implementation, the decay factors are stored in `A_vals`, and the state and output are updated via

```
x = A_vals * x + B * u[t],
```

followed by

```
y[t] = C @ x.
```

For the attention component, we build the causal mask $\mathsf{mask}_{t,s} = \mathbb{1}\{t \geq s\}$ for $t, s \in \{0, \ldots, T-1\}$ and then define the $T \times T$ kernel:

$$M_{t,s} = \sum_{n=1}^N C_n B_n \lambda_n^{t-s} \mathsf{mask}_{t,s}.$$

In the code, the kernel is constructed by summing

```
C[m] * B[m] * (A_vals[m] ** (t_idx - s_idx)) * mask,
```

over modes $m$. We then apply this kernel to the input via $y^{\mathrm{att}} = Mu$. Additionally, we compute the (i) generator rank, which is the numerical rank of the Vandermonde matrix with columns $t \mapsto \lambda_n^t$, and (ii) the numerical matrix rank of $M$ using `np.linalg.matrix_rank`.

**Baseline/Ground Truth.** We treat the SSM recurrence as a reference implementation of the dynamics and interpret any difference between $y$ and $y^{\mathrm{att}}$ as numerical error. For the rank analysis, theory predicts that for distinct decays $\lambda_1, \ldots, \lambda_N$ and sufficiently long sequences,

$$\mathrm{rank}(\mathrm{Vandermonde}(\lambda_1, \ldots, \lambda_N)) = N \quad \text{and} \quad \mathrm{rank}(M) = N.$$

Thus, we expect the generator rank and matrix rank $M$ to be equivalent to the state dimension $N$.

**Results.** For the default configuration $N = 2$, $\lambda = (0.5, 0.8)$, and 1000 random seeds, the maximum absolute error $\max_t |y_t - y_t^{\mathrm{att}}|$ is on the order of $10^{-14}$, which demonstrates that the sum of $N$ 1-SS masked attention heads numerically is equivalent to the diagonal SSM trajectory at the level of double-precision rounding error.

In all runs with distinct decays, the recorded generator rank and matrix rank satisfy:

$$\texttt{generator\_rank} = N, \quad \text{rank}(M) = N.$$

This confirms that the induced attention kernel has a semiseparable rank equal to the SSM state dimension. This numerically supports the diagonal SSM $\equiv$ sum-of-heads 1-SS attention correspondence in Section 3.1 and ties the number of decay modes to the rank of the resulting semiseparable attention operator.

**Time-Varying Extension.** We also verify that the diagonal SSM $\leftrightarrow$ sum-of-heads duality extends to the time-varying case mentioned in Section 3.1. We vary per-step parameters, $A_t = \text{diag}(\lambda_{t,1}, \ldots, \lambda_{t,N})$, $B_t, C_t \in \mathbb{R}^N$ and implement:

$$x_t^{(m)} = \lambda_{t,m} x_{t-1}^{(m)} + B_{t,m} u_t, \quad y_t = \sum_{m=1}^{N} C_{t,m} x_t^{(m)}.$$

Unrolling the recurrence gives:

$$y_t = \sum_{s \leq t} \sum_{m=1}^{N} C_{t,m} B_{s,m} \Big( \prod_{k=s+1}^{t} \lambda_{k,m} \Big) u_s,$$

which can be written as a sum of time-varying masked attention heads with kernels:

$$M_{t,s}^{(m)} = \Big( \prod_{k=s+1}^{t} \lambda_{k,m} \Big) \mathbb{1}\{t \geq s\}, \quad M_{t,s} = \sum_{m=1}^{N} C_{t,m} B_{s,m} M_{t,s}^{(m)}.$$

In our implementation, we construct $M$ by computing the products $\prod_{k=s+1}^{t} \lambda_{k,m}$ for each $(t, s, m)$ and then applying $y^{\text{att}} = Mu$. Across random draws of time-varying decays $A_t$, and time-dependent $B_t, C_t$, the maximum absolute error $\max_t |y_t - y_t^{\text{att}}|$ remains at the level of $10^{-14}$. This shows that the diagonal SSM is equivalent to sum-of-heads 1-SS duality in the time-varying setting.

### C.3 State Dimension vs. Semiseparable Rank (Section 3.2 and Theorem 3.1)

A key theoretical prediction is that the complexity of the attention kernel is governed by the state-space dynamics. To test this, we validate the relationship between the state dimension $N$ of a diagonal SSM and the semiseparable rank of the attention kernel, as stated in Section 3.2 and Theorem 3.1.

**Setup.** For a diagonal SSM with decays $\{a_m\}_{m=1}^{N}$ and scalar input $u_t$, the induced semiseparable attention kernel has entries:

$$M_{t,s} = \sum_{m=1}^{N} C_m B_m a_m^{t-s} \mathbb{1}\{t \geq s\},$$

where $B_m, C_m$ are fixed input/output weights. Theory predicts that, the semiseparable (generator) rank equals the number of distinct decays $\{a_m\}$ and is upper bounded by $N$, with repeated decays reducing the rank.

**Data/Constructions.** Our initial experiment fixes the sequence length $T = 15$ and considers four decay configurations:

- $N = 1$ with a single decay $\lambda = (0.9)$,

- $N = 2$ with distinct decays $\lambda = (0.5, 0.8)$,

- $N = 2$ with duplicated decays $\lambda = (0.7, 0.7)$,

- $N = 3$ with distinct decays $\lambda = (0.4, 0.6, 0.9)$.

For each case, we set $B = C = \mathbb{1} \in \mathbb{R}^N$ and work in double precision (float64). Additionally, we do a large sweep where we vary the sequence length $T \in \{10, 15, 20, 30, 40\}$, the random seeds, the state dimensions $N \in \{2, 3, 4, 5\}$, and decay sets.

**Model.** Given a decay vector $A_{\text{vals}} = (a_1, \ldots, a_N)$ and horizon $T$, we construct the standard indicator $\text{mask}_{t,s} = \mathbb{1}\{t \geq s\}$ together with time indices $t, s \in \{0, \ldots, T-1\}$. We then form the kernel:

$$M_{t,s} = \sum_{m=1}^{N} C_m B_m a_m^{t-s} \text{mask}_{t,s},$$

by looping over modes $m$ and summing $C_m B_m a_m^{t-s} \text{mask}_{t,s}$ into $M$. We also build a Vandermonde matrix (for generator rank computation):

$$G_{t,m} = a_m^t,$$

and compute its numerical rank. The generator rank approximates the semiseparable rank of the kernel as defined in the theory. The matrix rank of $M$ is obtained by applying `np.linalg.matrix_rank` directly to the $T \times T$ kernel.

**Baseline/Ground Truth.** The theoretical baseline is the statement that: (i) if all decays are distinct and $T \geq N$, then the generator matrix $G$ has full column rank $N$, and the induced kernel $M$ has rank $N$; (ii) if some decays are repeated, the generator rank drops to the number of distinct decays, and $M$ has the same reduced rank.

We treat the generator rank of $G$ as the "analytical" semiseparable rank, and use the matrix rank of $M$ as an independent numerical check that the constructed attention kernel demonstrates the low-rank structure predicted by the theory.

**Results.** For the four decay configurations, the experiment reports the state dimension $N$, the generator rank of $G$, and the matrix rank of $M$ for each case. In the $N = 1$ case with $\{0.9\}$, both the generator rank and matrix rank are 1. For $N = 2$ with distinct decays $\{0.5, 0.8\}$, both ranks are 2, matching the state dimension. When the decays are duplicated, $\{0.7, 0.7\}$, the generator rank and matrix rank drop to 1 despite $N = 2$, confirming that the second state contributes no new exponential mode. Finally, for $N = 3$ with distinct decays $\{0.4, 0.6, 0.9\}$, we obtain generator rank 3 and matrix rank 3.

These results are consistent across all tested configurations and numerically confirm Section 3.2 and Theorem 3.1: the semiseparable rank of the induced attention kernel equals the number of distinct decay modes and is bounded above by the SSM state dimension. Additionally, the large sweep over varying $T$, $N$, and random seeds consistently shows that the generator rank and matrix rank align as predicted.

## C.4 Time Complexity: $O(T)$ vs. $O(T^2)$ (Section 3.3)

While algebraically equivalent, the recurrent and attentional forms imply different computational costs. We quantify this trade-off by comparing the runtime scaling of the two implementations built from the same diagonal decays.

**Setup.** For fixed state dimension $N$ and feature dimension $d$, we measure wall-clock times of:

- A direct recurrence that updates the latent state in $O(T)$ time.

- An explicit attention implementation that materializes the full $T \times T$ kernel and applies it to the input in $O(T^2)$ time.

Our goal is to confirm that the measured wall-clock times scale linearly vs. quadratically with sequence length $T$.

**Data/Inputs.** For each sequence length $T \in \{150, 300, 600, 1200, 2400, 4800, 9600\}$, we draw an input matrix $U \in \mathbb{R}^{T \times d}$ with i.i.d. Gaussian entries $U_{t,j} \sim N(0, 1)$ with fixed seeds. Additionally, we vary the values $N \in \{2, 4, 8, 16\}$ and the feature dimension $d \in \{8, 16, 32, 64\}$ across different runs. The diagonal SSM decays are chosen as $N$ linearly spaced values in $[0.5, 0.8]$,

$$A_{\text{vals}} = (a_1, \ldots, a_N) = \text{linspace}(0.5, 0.8, N),$$

and are shared between the recurrence and attention implementations. All computations are performed in double precision (float64).

**Model.** The recurrence-based implementation a diagonal SSM with state dimension $N$ driven by a $d$-dimensional input sequence $U \in \mathbb{R}^{T \times d}$:

$$x_{t+1} = A_{\text{vals}} \odot x_t + BU_t,$$

where $x_t \in \mathbb{R}^N$ is the latent state at time $t$, $A_{\text{vals}} \in \mathbb{R}^N$ contains the diagonal entries, $B \in \mathbb{R}^{N \times d}$ is the input matrix, and $U_t \in \mathbb{R}^d$ is the $t$-th row of $U$.

The code initializes $x_0 = 0$ and iterates this update once per time step, resulting in an $O(TNd)$ algorithm that is linear in $T$ for fixed $N$ and $d$.

The attention-based implementation uses the same decay vector $A_{\text{vals}}$ to build the causal kernel $M \in \mathbb{R}^{T \times T}$ corresponding to the diagonal SSM. Using the causal mask $\text{mask}_{t,s} = \mathbb{1}\{t \geq s\}$ and time indices $t, s \in \{0, \dots, T-1\}$, it accumulates:

$$M_{t,s} = \sum_{m=1}^{N} a_m^{t-s} \mathbb{1}\{t \geq s\},$$

by looping over modes $m$ and adding $a_m^{t-s}\text{mask}_{t,s}$ into $M$. Subsequently, it applies this kernel to the input by multiplying $MU$. Algorithmically, building $M$ costs $O(NT^2)$ and the matrix-matrix product costs $O(T^2 d)$, so the overall complexity is quadratic in $T$ for fixed $N, d$.

**Measurement Protocol.** For each sequence length $T$ and fixed $(N, d)$, we generate a fresh input matrix $U$ and measure the wall-clock runtime of both implementations using /texttttime.perf_counter.

The reported recurrence time $t_{\text{rec}}$ and attention time $t_{\text{att}}$ are averaged over 3 repeats. The same decays $A_{\text{vals}}$ are used for both implementations.

**Results.** Figure 9 plots the measured runtime as a function of sequence length $T$ for both implementations, with solid lines showing the mean over 100 runs and shaded regions denoting $\pm$ one standard deviation. For $T \in \{150, 300, 600, 1200, 2400, 4800, 9600\}$, the recurrence time $t_{\text{rec}}$ grows approximately linearly with $T$, while the attention time $t_{\text{att}}$ grows superlinearly, following a quadratic trend. This empirically confirms the $O(T)$ vs. $O(T^2)$ scaling predicted by the diagonal SSM recurrence and its explicit attention dual, and it supports the efficiency discussion in Section 3.3.

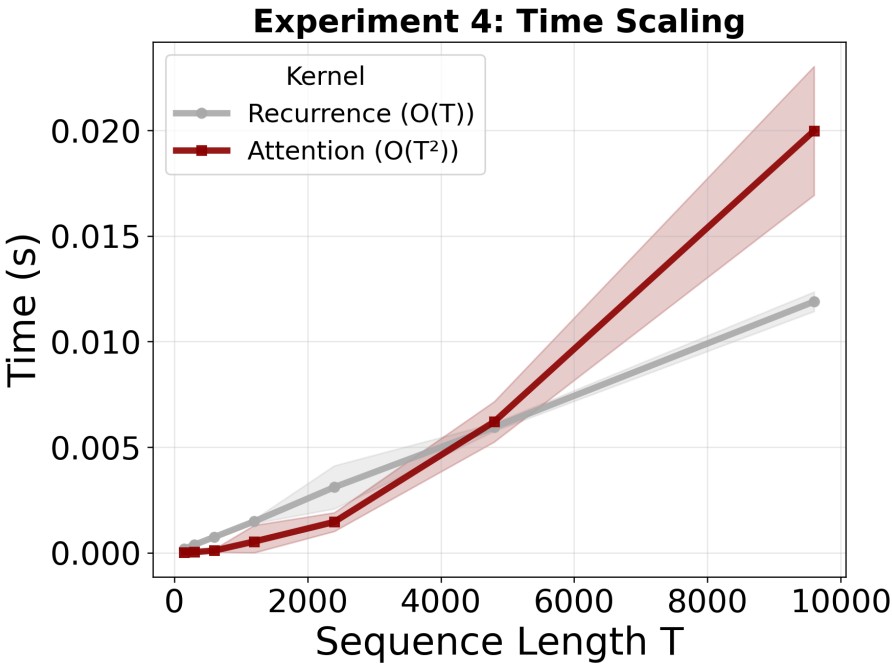

*Figure 9.* Wall-clock runtime vs. sequence length $T$ for the diagonal SSM implemented as a recurrence (gray, $O(T)$) and as explicit attention (red, $O(T^2)$). Curves show the mean over 100 runs; shaded regions denote $\pm$ one standard deviation. The recurrence scales linearly in $T$, while the attention implementation exhibits quadratic growth.

## C.5 Softmax Attention Rank Growth (Section 4)

For low-rank semiseparable structure, we study the numerical rank of causal softmax attention matrices built from random query/key vectors.

**Setup.** We validate the numerical rank of a causal softmax attention matrix built from random query/key vectors, as a "negative test" for low-rank semiseparable structure. While semiseparable kernels induced by diagonal SSMs are rank bounded by the number of distinct decay modes, Section 4 argues that generic softmax attention does not exhibit fixed low-rank structures. This experiment constructs full causal softmax attention matrices for increasing sequence lengths $T$ and measures their numerical matrix rank.

**Data/Inputs.** For each sequence length $T \in \{20, 40, 80, 160, 320, 640, 1280, 2560, 5120\}$, we sample a query matrix $Q \in \mathbb{R}^{T \times d_k}$ and key matrix $K \in \mathbb{R}^{T \times d_k}$ with i.i.d. Gaussian entries,

$$Q_{t,j}, K_{t,j} \sim N(0,1),$$

using NumPy `default_rng` with a fixed seed. We initially fix the key/query dimension to $d_k = 8$. No values $V$ are needed, and the experiment focuses solely on the attention weight matrix. All computations are performed in double precision (float64).

**Construction of Attention Kernel.** Given $Q, K \in \mathbb{R}^{T \times d_k}$, we build a causal softmax attention matrix $A \in \mathbb{R}^{T \times T}$ row by row. For each timestep $t \in \{0, \ldots, T-1\}$, we form unnormalized scores against all previous keys,

$$\mathsf{Score}_t[s] = \langle Q_t, K_s \rangle, \quad s = 0, \ldots, t,$$

and apply a numerically stable softmax,

$$\alpha_{t,s} = \frac{\exp(\mathsf{Score}_t[s] - \max_{r \leq t} \mathsf{Score}_t[r])}{\sum_{r \leq t} \exp(\mathsf{Score}_t[r] - \max_{r' \leq t} \mathsf{Score}_t[r'])}, \quad s \leq t.$$

In our implementation, this is implemented through `stable_softmax` and only the causal prefix $K[: t+1]$ is used.

The resulting weights populate the lower-triangular part of $A$:

$$A_{t,s} = \alpha_{t,s} \quad \text{for } s \leq t, \quad A_{t,s} = 0 \quad \text{for } s > t.$$

Thus, for each $T$ we obtain a full $T \times T$ causal attention matrix consistent with standard dot-product softmax attention, but with random queries and keys and no further structure imposed.

**Rank Estimation.** For each constructed attention matrix $A$, we compute its numerical rank using `np.linalg.matrix_rank`. The resulting integer $\mathrm{rank}(A) = \mathtt{matrix\_rank}(A)$ is recorded as the empirical rank for that sequence length. The experiment repeats this procedure independently for each $T$, using the same $d_k$ and random seed but fresh $Q$ and $K$ matrices.

**Results.** Figure 10 plots the numerical rank $\mathrm{rank}(A)$ as a function of sequence length $T$. The rank grows with $T$ and remains very close to the full-rank baseline $y = T$ for all sequence lengths considered, indicating that generic softmax attention is effectively full rank. The complementary view in Figure 11 shows the rank gap $T - \mathrm{rank}(A)$ on the same runs: the gap increases slowly with $T$ but stays tiny compared to $T$ (on the order of $10^2$ even when $T \approx 5{,}000$). Together, these plots confirm the qualitative claim of Section 4, that unstructured softmax attention does not admit a fixed low semiseparable rank, in contrast to the diagonal SSM kernels.

## C.6 Mamba Implementation

To contrast the structured SSM kernels with standard mechanisms, we analyze the numerical rank of generic softmax attention matrices as a "negative test" for low-rank semiseparable structure.

**Setup.** We base all sequence-modeling experiments on the minimal PyTorch implementation of Mamba by Ma (2023). It exposes a single Mamba block consisting of an input projection, depthwise convolution, selective SSM core, and output projection. We use this implementation as our *baseline Mamba* model and construct a matched *SSD-Mamba* variant by replacing the selective SSM core with our diagonal SSD-style SSM block from Appendix C.2, while keeping the surrounding

**Experiment 5: Softmax attention rank growth**

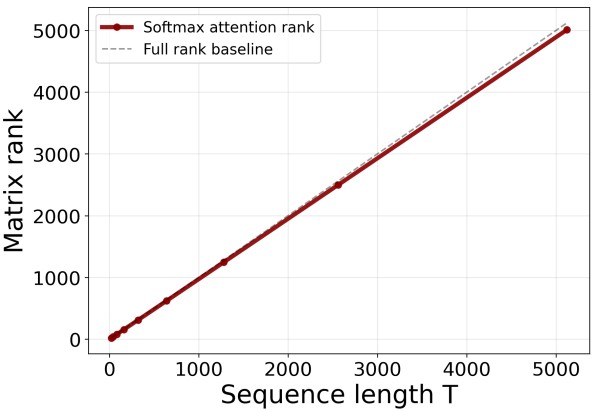

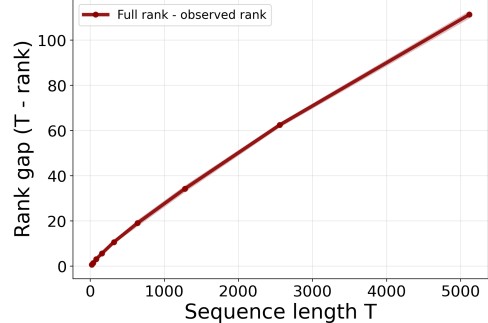

*Figure 10.* Experiment 5: numerical rank of the causal softmax attention matrix as a function of sequence length $T$. The solid curve shows the mean rank over random draws of $(Q, K)$ and the dashed line is the full-rank baseline $y = T$. The rank grows with $T$ and remains very close to full rank for all sequence lengths considered, indicating that generic softmax attention is effectively full rank.

*Figure 11.* Experiment 5: *rank gap* $T - \mathrm{rank}(A)$ for the same softmax attention matrices as in Figure 10. The gap grows slowly with $T$ but remains tiny compared to $T$ (on the order of $10^2$ even when $T \approx 5{,}000$), highlighting that generic softmax attention is effectively numerically full rank and does not admit a fixed low semiseparable rank.

architecture (embedding layer, convolution, SiLU nonlinearity, residual connections, and RMS normalization) identical. Both variants use the same hidden size $d_{\mathrm{model}}$, state dimension $d_{\mathrm{state}}$, expansion factor, number of layers, and vocabulary size.

We train all models from scratch using the AdamW optimizer with learning rate $1 \times 10^{-3}$, no learning-rate schedule, and default AdamW $\beta$ parameters. No additional regularization (dropout, weight decay beyond AdamW's default, or auxiliary losses) is applied in order to keep the comparison focused on the SSM block. We evaluate using cross-entropy loss and exact-match accuracy on the validation split, averaging results over multiple random seeds when indicated.

**Tasks/Datasets.** We consider a small-scale next-token prediction task on the WikiText-2 dataset (Merity et al., 2016). We use the `wikitext-2-raw-v1` release and its standard train/validation split. The corpus is tokenized at the word level using a simple whitespace tokenizer, and we cap the vocabulary at `max_vocab` tokens (in our experiments 20,000); all rarer tokens are mapped to a single `<unk>` symbol. From each split we construct rolling windows of length $L$ over the token stream and take the token following each window as the target (next-token at position $L$). In all runs we use $L = 128$. To keep experiments lightweight, we subsample 5,000 segments for training and 1,000 for validation in the default configuration.

**Models.** We train two Mamba-style architectures that only differ in their state-space block:

- **Mamba (selective SSM).** The reference model is the minimal PyTorch implementation of Mamba by Ma (2023), which we use unmodified. Each layer applies an input projection, a depthwise convolution, a SiLU nonlinearity, and then a selective state-space module with parameters $(A, B(x), C(x), \Delta(x), D)$. The matrix $A$ and skip connection $D$ are input-independent, while the step size $\Delta(x)$ and input/output vectors $B(x), C(x)$ are produced by learned linear maps of the current hidden state, and the state update is computed via the sequential `selective_scan` procedure from the original implementation.

- **SSD-Mamba (SSD-style diagonal SSM).** Our variant mirrors Ma's architecture but replaces the selective SSM core with the discrete diagonal SSD block described in Algorithm 1. In this block, each feature channel $k$ maintains an $N$-dimensional state $h_t^{(k)} \in \mathbb{R}^N$ with fixed, input-independent parameters $(a_{k,n}, b_{k,n}, c_{k,n}, d_k)$ stored in learnable tensors `A_log`, `B`, `C`, and `D`. The internal recurrence is

$$h_t^{(k,n)} = a_{k,n} h_{t-1}^{(k,n)} + b_{k,n} X_t^{(k)},$$

$$Y_t^{(k)} = \sum_{n=1}^{N} c_{k,n} h_t^{(k,n)} + d_k X_t^{(k)},$$

| Metric | Mamba (`mamba_simple`) | SSD-Mamba (`mamba_SSD_diag_exp`) |
|---|---|---|
| Train loss (start $\rightarrow$ end) | $38.9 \rightarrow 4.5$ | $27.1 \rightarrow 5.7$ |
| Train acc (end) | 33% | 17% |
| Val loss (start) | 18.1 | 16.8 |
| Best val loss | 10.5 | 9.7 |
| Val loss (end) | 11.7 | 9.8 |
| Best val acc | 9.5% | 11.1% |

*Table 1.* WikiText-2 next-token prediction results (averaged over six seeds) with $L = 128$, $d_{\text{model}} = 64$, $d_{\text{state}} = 16$, and two layers. The SSD-Mamba variant attains lower best validation loss and higher best validation accuracy than the baseline Mamba, despite fitting the training data less tightly.

implemented as an explicit diagonal scan over time, matching the SSD-style SSM in Algorithm 1. The surrounding input projection, depthwise convolution, SiLU nonlinearity, RMS normalization, and output projection are kept identical to the baseline Mamba, so differences in performance can be attributed to the choice of state-space parameterization.

**Training Details.** For WikiText-2 we train both models from scratch with AdamW (learning rate $1 \times 10^{-3}$, default $\beta$'s), batch size 64, and no learning-rate schedule or extra regularization beyond AdamW's built-in weight decay. All experiments use $d_{\text{model}} = 64$, $d_{\text{state}} = 16$, and 2 layers. We train for 10 epochs and report training and validation cross-entropy and accuracy. All weights are initialized randomly using PyTorch defaults (Gaussian initializations for linear layers); no pretrained components are used.

**Results on WikiText-2.** On the WikiText-2 next-token prediction task (sequence length $L = 128$, two layers, $d_{\text{model}} = 64$, $d_{\text{state}} = 16$), both models substantially reduce training loss but operate in a low-data, small-model regime.

Table 1 summarizes aggregate metrics averaged over six random seeds.

The baseline Mamba model (`mamba_simple`) fits the training set more aggressively, reaching lower training loss and higher training accuracy, but its validation loss improves over the first few epochs and then degrades, accompanied by a drop in validation accuracy toward the end of training, suggesting overfitting.

In contrast, the SSD-Mamba model (`mamba_SSD_diag_exp`) underfits slightly on the training set but shows more stable generalization, where the validation loss decreases more steadily and the best validation accuracy is higher than that of the baseline.

Overall, the SSD-style diagonal SSM block achieves better validation performance despite weaker training fit, which is consistent with the reduced overfitting suggested by Figure 12 and Figure 13.

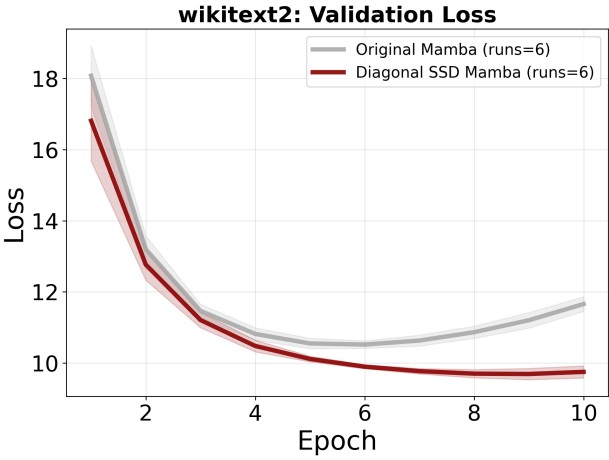

*Figure 12.* WikiText-2 validation loss over 10 epochs for the original Mamba model and the SSD-Mamba variant. Curves show mean $\pm$ one standard deviation over six random seeds.

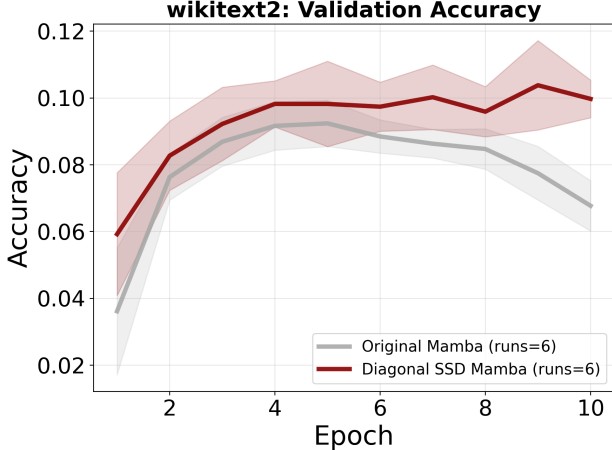

*Figure 13.* WikiText-2 validation accuracy over 10 epochs for the original Mamba model and the SSD-Mamba variant. Curves show mean $\pm$ one standard deviation over six random seeds.

## C.7 Time-Series Experiments: Diagonal SSM vs. Scalar SSM

This section provides a small time-series benchmark targeted to validate the sequence-processing task where a diagonal SSM with $N > 1$ state modes exhibits richer dynamics than a scalar SSM ($N = 1$) as described in Section 3.3. We use a synthetic regression problem built from a mixture of exponential decays and compare diagonal SSD-style SSM blocks with different state dimensions $N$.

**Setup.** We work with a 1D input sequence of length $T$ and train models to regress a fixed noisy target trace. The target $y_t$ is a sum of $M$ geometric decays with different rates,

$$y_t = \sum_{m=1}^{M} \alpha_m \lambda_m^t + \varepsilon_t, \qquad t = 0, \dots, T-1,$$

where $\lambda_m \in (0,1)$ are decay factors, $\alpha_m$ are coefficients, and $\varepsilon_t \sim \mathcal{N}(0, \sigma^2)$ is i.i.d. Gaussian noise. In the default configuration we use $M = 4$ modes with

$$(\lambda_1, \lambda_2, \lambda_3, \lambda_4) = (0.98, 0.94, 0.90, 0.80), \quad (\alpha_1, \alpha_2, \alpha_3, \alpha_4) = (1.0, 0.7, 0.5, 0.3),$$

sequence length $T = 200$, and noise level $\sigma = 10^{-2}$.

The input to the model is a scalar sequence $u_t \equiv 1$ for all $t$, so the entire task is to learn the temporal dynamics rather than the input content. We train using mean-squared error (MSE) between the predicted and target traces.

**Data.** For each run we generate a training set of $N_{\text{train}}$ sequences and a validation set of $N_{\text{val}}$ sequences, each sharing the same underlying noise-free trace but with independent noise:

$$y_t^{(i)} = \sum_{m=1}^{M} \alpha_m \lambda_m^t + \varepsilon_t^{(i)}, \qquad i = 1, \dots, N_{\text{train}} + N_{\text{val}}.$$

In the experiments shown in Figure 14 and Figure 15 we use $N_{\text{train}} \approx 10^3$, $N_{\text{val}} \approx 2.5 \cdot 10^2$, batch size 64, and work in double precision (float64) throughout.

**Model.** We use the same lightweight Mamba-style architecture as in the main text, but adapted for scalar time-series regression. A linear projection maps the scalar input to a hidden dimension $d_{\text{model}}$, followed by $L$ residual blocks; a final linear head maps back to $\mathbb{R}$. Each residual block consists of:

1. an input projection to $\mathbb{R}^{d_{\text{model}}}$,

2. a depthwise convolution and SiLU nonlinearity,

3. a state-space block,

4. RMS normalization and a residual connection.

For the state-space block we use the discrete diagonal SSD core from Algorithm 1, specialized to input-independent parameters. For each feature channel $k$ and mode $n \in \{1, \dots, N\}$ we learn decay, input, and output weights $(a_{k,n}, b_{k,n}, c_{k,n}, d_k)$ and update

$$h_t^{(k,n)} = a_{k,n} h_{t-1}^{(k,n)} + b_{k,n} X_t^{(k)}, \qquad Y_t^{(k)} = \sum_{n=1}^{N} c_{k,n} h_t^{(k,n)} + d_k X_t^{(k)},$$

where $X_t \in \mathbb{R}^{d_{\text{model}}}$ is the post-convolution feature vector at time $t$. The block is implemented as an explicit diagonal scan over time, matching Algorithm 1. We vary only the state dimension $N = d_{\text{state}}$ while keeping $d_{\text{model}}$, number of layers, optimizer, and all other hyperparameters fixed. The scalar SSM baseline is $N = 1$; diagonal multi-state models use $N > 1$.

**Training details.** All models are trained from scratch with AdamW, learning rate $1 \times 10^{-3}$, no learning-rate schedule, and default $\beta$ parameters. We train for 60 epochs per run with batch size 64 and report both the per-epoch training/validation MSE and the best validation MSE achieved during training. To estimate variability we repeat each configuration over multiple random seeds (for both parameter initialization and data noise) and aggregate metrics across seeds.

**Results.** Figure 14 shows the validation MSE curves over epochs, averaged across seeds, with shaded regions denoting $\pm$ one standard deviation. Both configurations quickly fit the mixture-of-decays signal and eventually reach very low MSE on this synthetic task, but the $N = 2$ diagonal SSM achieves lower validation loss and smaller variability across seeds in the early and middle stages of training.

Figure 15 shows the best validation MSE as a function of $N$. We see that increasing the number of diagonal modes from $N = 1$ to $N = 2$ yields a clear improvement in mean best-val MSE and a reduction in variance across runs, and in our runs additional modes (e.g., $N = 4$) do not harm performance. Taken together, these results indicate that adding a second diagonal state mode makes optimization easier and improves the best achievable error. This provides a concrete example of a diagonal SSM with multiple state modes can capture several distinct decay patterns more faithfully than a single-state SSM.

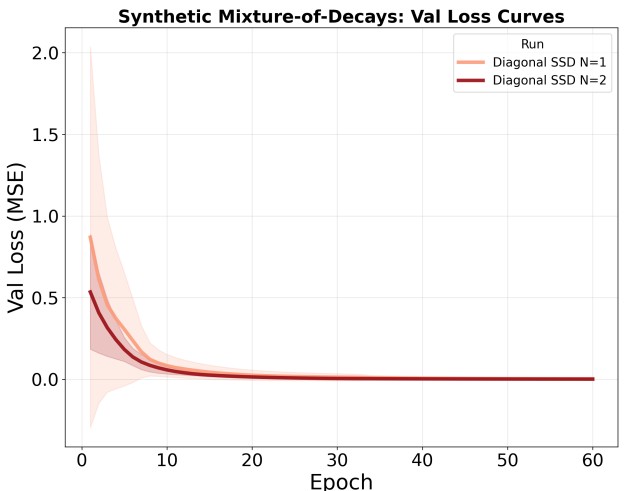

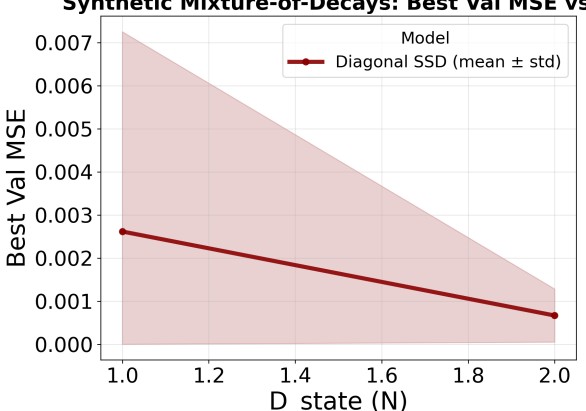

*Figure 14.* Validation MSE vs. epoch on the synthetic mixture-of-decays task. Curves show mean $\pm$ one standard deviation across random seeds for $N = 1$ and $N = 2$ diagonal SSD models. The $N = 2$ model converges faster and attains lower validation error.

*Figure 15.* Best validation MSE as a function of state dimension $N$ on the synthetic mixture-of-decays task. Each point aggregates multiple seeds (mean $\pm$ standard deviation). Increasing $N$ improves validation error relative to the scalar SSM ($N = 1$).

