# OpenReview forum: "On Structured State-Space Duality"
_ICML.cc/2026/Conference — ICML 2026 regular_

### Official Review · Reviewer_8h4e · 2026-03-05

**Soundness:** 3
**Presentation:** 3
**Significance:** 2
**Originality:** 3
**Overall Recommendation:** 4
**Confidence:** 3

**Summary:**

This work builds upon the state space duality (SSD) formulation, and extends it with general diagonal state matrices, showing they match scalar case complexity while supporting richer dynamics, and characterizes when SSMs are equivalent to semiseparable masked attention. As limitations, they show that the duality breaks down for standard softmax attention and for general SSMs, even with low state dimension.

**Compliance With Llm Reviewing Policy:**

Affirmed.

**Final Justification:**

The rebuttal reinforced my prior evaluation, I am keeping my positive score.

**Key Questions For Authors:**

The practical impact of these results remains unclear. In particular, when the authors refer to a broadened "design space for expressive yet efficient sequence modeling", what does this mean concretely? Could the authors provide examples or elaborate further?

**Limitations:**

Yes

**Strengths And Weaknesses:**

Strengths:
- Well-written and good quality paper
- Great contribution of extending SSD to a general diagonal form
- Nice self-contained and transparent work, which gives credit to previous work (Eidelman et al. 2014)
- Both theoretical results with limitations and empirical validations (in the Appendix) are presented

Weaknesses:
After carefully going through the paper, I cannot find substantial weaknesses. The one point that remains after reading the paper is that I cannot fully see where this work leads. More specifically, what is the practical impact of these new results? The authors wrote both in the abstract and in the conclusion that "Together, these results strengthen the theoretical bridge between recurrent SSMs and Transformer-style attention, and broaden the design space for expressive yet efficient sequence modeling". Could the authors elaborate more on this?

Small typos:
- Typo L36 right column "in in"
- L183 "on the right-hand side" - cannot find where it points

---

> ### Author Rebuttal · Authors · 2026-03-24
>
> > ### Weaknesses: After carefully going through the paper, I cannot find substantial weaknesses.
>
> Thanks for your encouraging words. We are very glad that you enjoyed the paper.
>
> > ### W1. The one point that remains after reading the paper is that I cannot fully see where this work leads. More specifically, what is the practical impact of these new results?
>
> In response, the most precise statement of our core contribution: this work precisely identifies where SSD extends beyond the scalar-identity regime, where it does not, and why.
>
> **The practical impact is model selection under exact guarantees.** Theorem 3.1 gives a necessary-and-sufficient condition for when a general SSM has a 1-SS masked-attention dual, while Section 4 and Proposition 4.1 identify failure cases, including standard softmax attention and even some low-dimensional general SSMs. **These results tell the audience which structured attention kernels can be converted exactly into recurrent realizations, and which ones provably cannot.**
>
> We remind the reviewer that, the submission also already contains a concrete matched numerical justifications.  Appendix C.6 replaces the selective SSM core in a Mamba-style block with the diagonal SSD core from Algorithm 1. Table 1 reports lower best validation loss (9.7 vs. 10.5) and higher best validation accuracy (11.1% vs. 9.5%) without hardware-specific kernels.
>
> > ### W2. what is the practical impact of these new results?
>
> In response, we here highlight three concrete impacts already established in the paper:
>
> - First, **Theorem 3.1 gives an exact structural criterion for when a masked-attention kernel has an exact recurrent SSM dual**: the induced $N$-SS kernel must decompose into diagonal blocks, each with at most $N$ new columns. Conversely, Section 4 shows that exact finite-state duals do not exist for standard softmax attention, and may fail even for low-dimensional general SSMs. So the paper turns this question from heuristic intuition into a precise yes/no characterization.
>
> - Second, **the paper proves a strict representational gain at fixed state size.** Remark 3.1 gives an explicit $N=2,\ T=4$ kernel realizable by a 2-dimensional diagonal SSM but by no scalar-identity SSM with the same $N$. So diagonal SSD is not a reparameterization of scalar SSD. It adds exact multi-timescale kernels while preserving recurrent $O(NTd)$ complexity (Section 3.3).
>
> - Third, **Appendix C shows that this extra flexibility can matter empirically.** In Appendix C.6, replacing the state-space core in a matched Mamba-style model with the diagonal SSD block improves best validation loss from 10.5 to 9.7 and best validation accuracy from 9.5% to 11.1% on WikiText-2 (Table 1), albeit in a small-scale setting and without specialized kernels. In Appendix C.7, increasing the number of diagonal modes from $N=1$ to $N=2$ reduces best validation MSE and variance on a mixture-of-decays task (Figures 13–14). Figure 8 also verifies the expected linear-versus-quadratic scaling of recurrence versus explicit attention.
>
> > ### W3. typos
>
> Thanks for careful proofreading! We have fixed these in the final version.
>
>
> > ### Q1. The practical impact of these results remains unclear. In particular, when the authors refer to a broadened "design space for expressive yet efficient sequence modeling", what does this mean concretely? Could the authors provide examples or elaborate further?
>
> To clarify, here “efficient” means exact recurrent $O(NTd)$ realizability, not a presently optimized GPU kernel (Section 3.3 & Remark 3.3). “Strengthen the theoretical bridge” means the paper extends the exact correspondence from scalar SSD (Proposition 2.1) to diagonal SSMs (Sections 3.1-3.2), then characterizes the general SSM class that still has a 1-SS masked-attention dual (Theorem 3.1), and finally proves where that bridge stops (Section 4).
>
> Concretely, the designer is no longer restricted to a single shared decay trajectory $a_t $. Section 3.1 shows that a diagonal SSM yields a sum of $N$ independent 1-SS components, i.e., $N$ separate decay modes. Section 3.2 then shows that, when the diagonal state matrices are full rank, these modes still admit a single 1-SS masked-attention representation after reparameterizing $B$ and $C$. This expands the exact linear-time family from single-mode scalar SSD to multi-mode diagonal SSD.
>
> Also, this expansion is strict. Remark 3.1 constructs a concrete kernel with 3 new columns that is realizable by a diagonal SSM with $N=2$ but impossible for any scalar-identity SSM with the same state dimension. Appendix C.7 gives the corresponding modeling example: multiple diagonal modes fit mixtures of distinct decay rates more faithfully than a single mode.
>
> So “broaden the design space” means: **more exact linear-time kernels are now available to the designer, and the frontier of that availability is explicit rather than guessed.** We will make these concrete examples explicit in the abstract and conclusion.

---

> > ### Author Rebuttal · Reviewer_8h4e · 2026-04-02
> >
> > I thank the authors for their detailed rebuttal and for addressing my concerns regarding practical impact with concrete examples. I am keeping my positive assessment and score.

---

> > > ### Author Response · Authors · 2026-04-03
> > >
> > > We are very glad to see our rebuttal meets your expectations! Thank you again for your review and kind words! Wish you a wonderful day!

---

### Official Review · Reviewer_aLyn · 2026-03-09

**Soundness:** 3
**Presentation:** 3
**Significance:** 3
**Originality:** 3
**Overall Recommendation:** 4
**Confidence:** 3

**Summary:**

This paper studies the theoretical foundations of Structured State-Space Duality (SSD), connecting sequence modeling via state-space models (SSMs) to structured masked attention (SMA) layers. The authors extend the characterization of SSM layers beyond the scalar case to diagonal and general SSMs. Specifically, the paper establishes a sufficient condition for diagonal SSMs to be equivalent to 1-semiseparable (1-SS) SMA, with asymptotic bounds on algorithmic FLOP complexity. Furthermore, the authors provide a necessary and sufficient condition for a general SSM to admit a 1-SS masked-attention dual, formulated in terms of the structure of its associated N-semiseparable matrix.

**Compliance With Llm Reviewing Policy:**

Affirmed.

**Final Justification:**

The paper provides a step forward in developing the SSD theory with several new results on diagonal and general SSMs in the SSD framework.While the rebuttal did not fully resolve all of mine concerns, the authors did respond to all of my questions and I believe that the work may value the community in current form as well, which is why I maintained my original positive score.

**Key Questions For Authors:**

- In Theorem 3.1, what is the meaning of "several diagonal blocks"?
- Minor Corrections: There is a duplicated "in" on Line 43: "We provide a self-contained proof in in the SSM-attention language..."

**Limitations:**

yes

**Strengths And Weaknesses:**

#### Strengths
- The paper is well-structured and logically organized. Each section has a clearly defined scope, followed directly by the corresponding claims, proofs, and explanations.
- The paper provides a clear structural intuition for diagonal SSMs, elegantly connecting them to a sum of 1-SS SMAs in the general case, and proves a necessary condition for equivalence to the known 1-SS SMA case. This yields deeper insights into SSD for more complex SSMs and strengthens the theoretical connection between SMA and SSM layers.
- A precise characterization of the equivalence between general SSMs and 1-SS SMAs is a meaningful theoretical contribution that provides a solid mathematical foundation to guide future research on the SSD framework.
#### Weaknesses
- As the authors explicitly acknowledge, much of the underlying mathematical tools concerning semiseparable and sequentially semiseparable (SS/SSS) matrices are already established in the structured-matrix literature. Consequently, the novelty of this work lies more in reformulating and integrating these concepts into the machine learning context, rather than introducing entirely new mathematics.
- While it is reasonable that a theoretical paper leaves the development of a hardware-optimized implementation for diagonal SSDs to future work, relying solely on an asymptotic FLOP analysis somewhat misses the primary motivation of the original SSD framework. A central motivation in the SSD (e.g., in Mamba-2) is improving training efficiency through better utilization of modern hardware optimized for matrix multiplications (e.g., Tensor Cores). Demonstrating optimal asymptotic FLOPs does not fully capture this advantage.
- There is a heavy reliance on inline equations and dense mathematical definitions that sometimes break awkwardly across newlines, making the text difficult to parse.

---

> ### Author Rebuttal · Authors · 2026-03-24
>
> > ### W1. As the authors explicitly acknowledge, much of the underlying mathematical tools concerning semiseparable and sequentially semiseparable (SS/SSS) matrices are already established in the structured-matrix literature. Consequently, the novelty of this work lies more in reformulating and integrating these concepts into the machine learning context, rather than introducing entirely new mathematics.
>
> Thanks for pointing this out. Yet, we remind the reviewer that, **the only overlap with literature is the classical SS/SSS equivalence itself. This is already clarified in Remark 3.5. All other results are genuinely new and novel.** We believe it’s not fair to reduce the whole paper to reformulation.
>
> To clarify, Our Remark 3.5 already states that the semiseparable/quasiseparable equivalence predates this paper, and that the concerned contribution (Proposition 3.1) there is a self-contained, constructive, strictly causal proof in SSM/SSD notation. This result is required to be used as the basis of the later diagonal and general-duality results.
>
> Under this basis, all these results are new:
> - Sections 3.1–3.2 derive the diagonal-SSM dual constructively and, under $\det(A_t)\neq 0$, collapse it to the single 1-SS form $M=1SS(1,\ldots,1)\odot(B'C'^\top)$.
> - Section 3.3 gives Algorithm 1 and the $O(NTd)$ complexity analysis.
> - Section 3.4 gives the exact structural criterion for when a general SSM has a 1-SS dual (Proposition 3.2, Lemma 3.1, Theorem 3.1).
> - Section 4 adds the negative result for softmax via rank explosion.
>
> To further improve clarity and avoid confusion, we will highlight the caveat now stated in Remark 3.5 into the introduction in the final version.
>
> ---
>
> > ### W2. While it is reasonable that a theoretical paper leaves the development of a hardware-optimized implementation for diagonal SSDs to future work, relying solely on an asymptotic FLOP analysis somewhat misses the primary motivation of the original SSD framework. A central motivation in the SSD (e.g., in Mamba-2) is improving training efficiency through better utilization of modern hardware optimized for matrix multiplications (e.g., Tensor Cores). Demonstrating optimal asymptotic FLOPs does not fully capture this advantage.
>
> Yes, you are absolutely correct that asymptotic FLOP optimality does not by itself demonstrate tensor-core speedups. However, we do not claim such a systems result.
>
> In fact, our Remark 3.3 states this explicitly: no specialized diagonal-SSD kernel is provided, and hardware-aware implementation is future work. **The scope of the paper is instead to characterize where SSD extends and where it provably fails**: constructive diagonal duality and its $O(TNd)$ recurrence (Sections 3.1-3.3, Algorithm 1), an exact criterion for 1-SS duality of general SSMs (Section 3.4, Theorem 3.1), and impossibility results for broader extensions (Section 4). Appendix C.2–C.4 numerically confirms the predicted equivalence, rank, and scaling behavior.
>
> Thus the absence of a hardware-tuned kernel narrows implementation scope, but does not weaken the paper’s claimed contribution.
>
> ---
>
> > ### W3. There is a heavy reliance on inline equations and dense mathematical definitions that sometimes break awkwardly across newlines, making the text difficult to parse.
>
> Thanks for the comment. Yes, this makes the paper dense to read. We will relax the space in the final version through the extra page.
>
> ---
>
> > ### Q1. In Theorem 3.1, what is the meaning of "several diagonal blocks"?
>
> Thanks for the question. Here **“several diagonal blocks” means a union of contiguous principal diagonal blocks containing all nonzero mass, not an arbitrary block decomposition.**
>
> This was made precise  in the paper, Lemma 3.1, Part 2: there exist indices $1=t_0<t_1<\cdots<t_k=T+1$ such that all nonzero entries of $M$ lie in the contiguous principal blocks $M_{t_{i-1}:t_i,;t_{i-1}:t_i}$, i.e. on intervals $[t_{i-1},t_i-1]$ along the diagonal; off-block entries are zero.
>
> Thus, in the 1-SS mask $L=1SS(p)$, the block boundaries are exactly the zero locations of $p_2,\ldots,p_T$, because those zeros make every cross-boundary product vanish.
>
> To revise, we will replace the phrase in Theorem 3.1 with this interval notation in the final version.
>
> ---
>
> > ### Q2. Minor Corrections: There is a duplicated "in" on Line 43: "We provide a self-contained proof in in the SSM-attention language..."
>
> Thanks for pointing this out. We have fixed it accordingly!
>
> ---
>
> Thank you for your time and valuable feedback. Please do not hesitate to let us know if there are any other aspects of our work that you would like us to clarify.

---

> > ### Author Rebuttal · Reviewer_aLyn · 2026-04-04
> >
> > All of my questions have been addressed, I am keeping my score.

---

> > > ### Author Response · Authors · 2026-04-04
> > >
> > > Thank you again for your kind review and constructive comments. Happy to see our rebuttal was satisfactory. We will relax the spacing from inline math to improve readability as promised. Wish you a wonderful day!

---

### Official Review · Reviewer_iQTD · 2026-03-12

**Soundness:** 3
**Presentation:** 3
**Significance:** 3
**Originality:** 3
**Overall Recommendation:** 4
**Confidence:** 3

**Summary:**

The paper extends structured state-space duality (SSD) beyond the scalar-times-identity setting to a broader diagonal SSM class. It derives an attention-like dual for diagonal SSMs, gives an explicit diagonal SSD algorithm, characterizes when general SSMs admit a 1-semiseparable masked attention dual, and presents negative results showing that the duality does not naturally extend to softmax attention or to all low-rank / low-dimensional SSMs.

**Compliance With Llm Reviewing Policy:**

Affirmed.

**Final Justification:**

The rebuttal have addressed my main concerns, so I decide to increase the score.

**Key Questions For Authors:**

1. Which results are genuinely new relative to prior SSD work and the structured-matrix / quasiseparable literature, and which are better viewed as reformulations or specialized derivations?
This would directly affect my originality assessment.

2. Can the authors provide a more operational, parameter-level interpretation of the 1-SS dual characterization?
If the result can be made more constructive or design-oriented, I would view the paper as more significant.

3. Can the authors better justify the practical relevance of diagonal SSD beyond toy runtime and small-scale modeling results?
Stronger evidence on more realistic sequence modeling settings could improve my significance score.

4. Would the authors consider tightening the paper’s positioning so that it is more explicitly framed as a structural/theoretical contribution?
A more precise positioning would make me more comfortable with the paper overall.

**Limitations:**

yes

**Strengths And Weaknesses:**

- Strengths: The paper is focused and technically coherent. I particularly appreciate that it does not only present positive constructions, but also gives structural limitations, which makes the overall contribution feel more complete. The characterization of when a general SSM admits a 1-SS masked attention dual is, to me, the strongest part of the submission.

- Weaknesses: My main concern is novelty positioning. Parts of the mathematical structure appear related to existing semiseparable / quasiseparable matrix literature, and the paper could do more to distinguish genuinely new results from SSD-specific reformulations of prior structural facts. My second concern is significance: the empirical section is useful as a sanity check, but currently too limited to establish a strong practical case for why this broader diagonal SSD materially expands the usable design space of modern sequence models.

---

> ### Author Rebuttal · Authors · 2026-03-25
>
> > ### W1. ... distinguish genuinely new results verse SSD-specific reformulations of prior structural facts.
>
> Thanks for pointing this out. To clarify, **the concerned overlapping is *only* the self-contained, constructive, strictly causal proof in SSM/SSD notation (Proposition 3.1).** This result is required to be used as the basis of the later diagonal and general-duality results.
>
> **All other results are genuinely new** (Sec 3.1-3.4 and Sec 4):
> - the diagonal-SSM attention-like dual, the explicit full-rank diagonal $1$-SS construction,
> - Algorithm 1 with the matching $O(NTd)$ complexity statement,
> - the necessary-and-sufficient criterion in Proposition 3.2 / Lemma 3.1 / Theorem 3.1, and
> - the negative limits for softmax attention and general low-dimensional SSMs.
>
> Sorry for any confusion caused. We will revise the introduction to state this separation earlier and more bluntly in the final version.
>
> > ### W2. My second concern is significance: the empirical section is useful as a sanity check, but currently too limited to establish a strong practical case for why this broader diagonal SSD materially expands the usable design space of modern sequence models.
>
> Sorry for any confusion caused. But we believe there might be a potential overlook. As Remark 3.3 states, **the goal is to characterize the duality rather than provide optimized implementations.** The significance claim is very specific: diagonal SSD extends SSD beyond the scalar-times-identity regime while retaining the same asymptotic time complexity and parallelization structure. **We do not mean to establish a strong practical case. What's established here is a rigorous opening of such avenue.** As mentioned throughout, we leave the materialization of usable hardware-speedup design for future work.
>
> So, our Appendix C is therefore targeted, not incidental:
> - C.1-C.5 numerically validate the stated equivalences and limits;
> - C.6 replaces only the SSM core in a matched Mamba-like model on WikiText-2, and
> - Table 1 reports better best validation loss and accuracy for the SSD-Mamba variant;
> - C.7 isolates the multi-mode benefit on a mixture-of-decays task.
>
> It is designed to validate the main consequences of the theory rather than serve as a full systems benchmark. We hope this clarification has addressed your concern.
>
> > ### Q1. Which results are genuinely new relative to prior SSD work and the structured-matrix / quasiseparable literature, and which are better viewed as reformulations or specialized derivations? This would directly affect my originality assessment.
>
> To clarify:
>
> - **Reformulation:** Proposition 3.1, together with Remark 3.5, is a constructive SSD-language proof of a known structural equivalence.
>
> - **New results:** all other results. Please also see above response to W1.
>
> **We believe these results remain novel even after fair and factual comparison against literature.** Thank you for the question!
>
> > ### Q2. Can the authors provide a more operational, parameter-level interpretation of the 1-SS dual characterization? If the result can be made more constructive or design-oriented, I would view the paper as more significant.
>
> We respectfully remind the reviewer that, for the diagonal full-rank case, **the paper already gives a parameter-level construction.**
>
> Specifically, Section 3.2 rewrites each diagonal mode via $b_t^{\prime n} := b_t^n \prod_{r\le t} A^r_{nn}$ and $c_t^{\prime n} := c_t^n / \prod_{r\le t} A^r_{nn}$, yielding $M = 1SS(1,\dots,1)\odot(B'C'^\top)$. This is constructive, not merely existential.
>
> For general SSMs, Theorem 3.1 is intentionally structural rather than a training recipe: Definition 3.2 interprets a “new column” as a new independent memory direction. Also, Theorem 3.1 says a $1$-SS dual exists iff each active diagonal block introduces at most $N$ such directions.
>
> **In operational terms, a $1$-SS dual is possible exactly when the kernel’s memory growth never exceeds the state budget within any nonzero block.**
>
> We understand our current draft is not clear enough to highlight these. We will revise accordingly in the final version.
>
> > ### Q3. Can the authors better justify the practical relevance of diagonal SSD beyond toy runtime and small-scale modeling results? Stronger evidence on more realistic sequence modeling settings could improve my significance score.
>
> We agree that broader experiments on more realistic workloads would strengthen a stronger practical claim. Yet, that is not the claim of this paper. Due to space limits here, please also see our above W2 response.
>
> > ### Q4. Would the authors consider tightening the paper’s positioning so that it is more explicitly framed as a structural/theoretical contribution? A more precise positioning would make me more comfortable with the paper overall.
>
> Thanks for the suggestion. Agreed and will do.
>
> ---
>
> Thanks for your time and effort. We have taken all comments into careful consideration. We look forward to further feedback and discussion.

---

> > ### Author Rebuttal · Reviewer_iQTD · 2026-04-04
> >
> > Thanks for the author‘s candid rebuttal. I have no concerns.

---

> > > ### Author Response · Authors · 2026-04-04
> > >
> > > Thank you again for your detailed review and kind words! We are very happy to see our rebuttal meets your expectations. Wish you a wonderrful day!

---

### Official Review · Reviewer_xcvz · 2026-03-16

**Soundness:** 4
**Presentation:** 3
**Significance:** 3
**Originality:** 4
**Overall Recommendation:** 4
**Confidence:** 3

**Summary:**

This paper theoretically studies structured state-space duality (SSD) in depth and extends SSD from the previously considered scalar-identity state matrices to diagonal state matrices. The authors further show that diagonal SSMs have the same computational complexity as scalar-identity SSMs. In addition, the paper provides a necessary and sufficient condition for an SSM to admit a 1-semiseparable masked attention dual.

**Compliance With Llm Reviewing Policy:**

Affirmed.

**Final Justification:**

All of my questions and concerns have been addressed. I keep my positive rating.

**Key Questions For Authors:**

1. Why is the duality with 1-SS masked attention beneficial? What practical implications does 1-SS masked attention have for applications or implementations?

2. Although the effectiveness of diagonal SSMs in time-series regression has been validated, what is the direct connection between "richer dynamics" and improved numerical performance? Do the authors mean that diagonal SSMs can better capture high-frequency variations in time series?

3. What is the intuition behind Theorem 3.1 that could inspire a follow-up method? A few examples or illustrative figures may help clarify this point.

**Limitations:**

Yes

**Strengths And Weaknesses:**

**Strengths:**

- This paper provides detailed formulations and proofs that further complete the duality between SSMs and masked attention. The theoretical formulation presented in the paper may pave the way for future work to further investigate this duality.

**Weaknesses:**

- One motivation for establishing the duality between SSMs and masked attention is to leverage the strong kernel-level parallelism of SSM algorithms. However, diagonal SSMs may face challenges in efficient implementation.

- Although this work is primarily theoretical, the current numerical experiments appear somewhat limited. Additional experiments on established benchmarks and tasks could further justify the advantages of diagonal SSMs.

---

> ### Author Rebuttal · Authors · 2026-03-24
>
> > ### W1. One motivation for establishing the duality between SSMs and masked attention is to leverage the strong kernel-level parallelism of SSM algorithms. However, diagonal SSMs may face challenges in efficient implementation.
>
> Yes it’s true we do not provide a specialized high-performance kernel. Yet, we remark that, this is due our scope definition, explicitly acknowledged. We do not mean to establish a strong practical, efficient implementation. What's established here is a provably correct opening of such avenue.
>
> Specifically, Section 3.3 proves the algorithmic claim we actually make: Algorithm 1 computes diagonal SSD in $O(NTd)$ FLOPs, with $O(NTd)$ total memory, and its first three stages parallelize over $n\in[N]$ and then over feature channels. Already in the paper, Remark 3.3 states that no specialized kernel currently accelerates diagonal SSD and that optimized implementations are future work.
>
> Please also see our response to W2 of `iQTD `. Thank you!
>
> ---
>
>
> > ### W2. Although this work is primarily theoretical, the current numerical experiments appear somewhat limited. Additional experiments on established benchmarks and tasks could further justify the advantages of diagonal SSMs.
>
> Thanks for the question. Yet we believe this deviates from our scope and presented appendix. This work precisely identifies where SSD extends beyond the scalar-identity regime, where it does not, and why. And our numerical results are meant to support our results under this scope.
>
> Therefore, while it’s true the benchmark suite is not broad enough to support a general empirical-superiority claim, our submission did not really make that stronger claim.
>
> ---
>
> > ### Q1. Why is the duality with 1-SS masked attention beneficial? What practical implications does 1-SS masked attention have for applications or implementations?
>
> The benefit is exact recurrent realizability.
>
> Proposition 2.1 shows that the same sequence operator can be written either as masked attention $Y=(M\odot(CB^\top))X$ or as a linear-time recurrence. Section 3.2 strengthens this for full-rank diagonal SSMs by rewriting their kernel as a single 1-SS mask times $B'C'^\top$, and Theorem 3.1 gives the exact structural condition more generally.
>
> The practical implication is not that explicit 1-SS attention is itself the target architecture. It is that any operator with this structure admits an exact finite-state $O(T)$ realization. This is verified in Appendix C.4: then contrasts that recurrent realization with the $O(T^2)$ explicit attention form.
>
> ---
>
>
> > ### Q2. Although the effectiveness of diagonal SSMs in time-series regression has been validated, what is the direct connection between "richer dynamics" and improved numerical performance? Do the authors mean that diagonal SSMs can better capture high-frequency variations in time series?
>
> Thanks for pointing this out. Not really. We did not claim any generic high-frequency advantage.
>
> In fact, our claim here is very specific: diagonal SSMs realize a strictly larger kernel class than scalar-identity SSMs at the same state dimension.
>
> For details, Section 3.1 decomposes a diagonal SSM into a sum of $N$ independent 1-SS components. Remark 3.1 then gives an explicit $N=2$, $T=4$ kernel that a diagonal SSM realizes but any scalar-identity SSM with the same $N$ cannot, because that kernel has three new columns. Appendix C.3 confirms that distinct decays add independent modes and rank, while Appendix C.7 shows better regression performance when the target is a mixture of multiple decay modes.
>
> **Thus, the direct link is multiple independent decay/time-scale modes, not specifically high-frequency behavior.**
>
> ---
>
>
> > ### Q3. What is the intuition behind Theorem 3.1 that could inspire a follow-up method? A few examples or illustrative figures may help clarify this point.
>
> **The intuition is "column growth equals memory growth."**
>
> Definition 3.2 calls a column “new” when it is not in the span of earlier lower-triangular columns. In state-space terms, that is a new independent memory direction. Proposition 3.2 proves that a fine 1-SS masked-attention factorization exists iff a block has at most $N$ new columns, Lemma 3.1 extends this blockwise, and Theorem 3.1 lifts the statement to general SSMs.
>
> So the theorem says exactly when an $N$-dimensional state can carry all causal directions required by the kernel. In our final version, we will highlight this “memory-direction” explanation next to Theorem 3.1 and point readers to Remark 3.8.
>
> **A plausible follow-up is to parameterize or regularize kernels so that each diagonal block introduces few new columns.** What is not currently established is which such parameterization is best in practice. And yes, we will add few illustrative figures to make the intuition clearer in our final version!
>
> ---
>
> We hope that our responses adequately address the reviewer's concerns, and we look forward to any further feedback. Thank you for your time and consideration.

---

> > ### Author Rebuttal · Reviewer_xcvz · 2026-04-03
> >
> > All of my questions and concerns have been addressed. I will keep my positive rating.

---

> > > ### Author Response · Authors · 2026-04-04
> > >
> > > We are very glad that our rebuttal was satisfactory! Thank you again for your constructive comments. They are very helpful for us to improve the draft. Wish you a wonderful day!

---

### Decision · Program_Chairs · 2026-04-30

**Decision:**

Accept (regular)

**Comment:**

The paper provides an important theoretical contribution to the SSM field by extending structured state-space duality to diagonal SSMs. The majority of the issues raised by the reviewers have been resolved during the rebuttal, and as a result, all reviewers vote for acceptance. Thus, I recommend accepting the paper.